# Control of mouse limb initiation and antero-posterior patterning by Meis transcription factors

Irene Delgado [1], Giovanna Giovinazzo[1], Susana Temiño[1], Yves Gauthier[2], Aurelio Balsalobre[2], Jacques Drouin[2] & Miguel Torres [1✉]

Meis1 and Meis2 are homeodomain transcription factors that regulate organogenesis through cooperation with Hox proteins. Elimination of Meis genes after limb induction has shown their role in limb proximo-distal patterning; however, limb development in the complete absence of Meis function has not been studied. Here, we report that *Meis1/2* inactivation in the lateral plate mesoderm of mouse embryos leads to limb agenesis. Meis and Tbx factors converge in this function, extensively co-binding with Tbx to genomic sites and co-regulating enhancers of *Fgf10*, a critical factor in limb initiation. Limbs with three deleted Meis alleles show proximal-specific skeletal hypoplasia and agenesis of posterior skeletal elements. This failure in posterior specification results from an early role of Meis factors in establishing the limb antero-posterior prepattern required for *Shh* activation. Our results demonstrate roles for Meis transcription factors in early limb development and identify their involvement in previously undescribed interaction networks that regulate organogenesis.

[1] Cardiovascular Development Program, Centro Nacional de Investigaciones Cardiovasculares (CNIC), Madrid, Spain. [2] Laboratoire de Génétique Moléculaire, Institut de Recherches Cliniques de Montréal, Montreal, QC, Canada. ✉email: mtorres@cnic.es

Limbs initiate as swellings from the lateral body wall that form by the localized epithelial-to-mesenchymal transition of the somatic coelomic epithelium[1]. Limb induction requires *Tbx4/5* and *Fgf10* activation in the lateral plate and their elimination produces limbless embryos[2]. *Tbx5* in the forelimb and *Tbx4* in the hindlimb is directly regulated by several inputs, including nuclear ßcatenin, retinoic acid, and specific combinations of axial Hox expression[3–5]. In cooperation with persistent nuclear ßcatenin, Tbx factors, in turn, induce limb-bud mesenchyme to express *Fgf10*. Fgf10 signaling to the overlying surface ectoderm activates several FGF and Wnt factors, which signal back to the mesenchyme to maintain *Fgf10* expression, forming a positive-feedback loop essential for limb-bud establishment, growth, and distalization[6–9].

Progression along the proximo-distal (PD) axis is concomitant with limb patterning along the antero-posterior (AP) axis. The main regulator of limb AP patterning is the secreted factor Shh, which is expressed from the zone of polarizing activity (ZPA), a discrete mesenchymal region at the posterior end of the limb bud[10,11]. AP prepatterning of the limb-forming region is required for *Shh* activation and starts simultaneously with limb induction, much earlier than the onset of *Shh* expression. The posterior field of the limb bud is established by AP-regulated expression of axial Hox9 paralogs in the forelimb and *Islet1* in the hindlimb, which induce posteriorly restricted expression of *Hand2*[12,13]. Hand2, in turn, restricts *Gli3* mRNA expression to the anterior region of the limb bud, producing its first AP compartmentalization[2,14]. *Hand2* is induced not only by Hox9 paralogs, but also by Hoxd13, which binds to the *Hand2* locus[15], suggesting direct regulation. At later stages, *Shh* expression is induced in the ZPA through cooperation between Hand2[16–18] and posterior HoxD proteins[19–23].

Meis1 and 2 belong to the TALE (three amino-acid loop extension) homeodomain transcription factor family and regulate Hox activity by interacting directly with posterior Hox proteins[24,25] and indirectly with anterior Hox proteins through Pbx, another TALE homeodomain family member[26]. Meis factors associate extensively but not exclusively with Hox-binding sites in the genome and also act as cofactors for non-Hox transcription factors[27–30]. Meis1 and 2 are homogeneously expressed in the lateral plate mesoderm from pre-limb stages and, following limb initiation, become expressed in a proximal-to-distal gradient in the growing limb bud[31]. Meis overexpression inhibits limb-bud

distalization[32–34], whereas *Meis* elimination after limb induction results in distalization and strong disruption of proximal segments[31]. How limbs develop in the complete absence of Meis expression in the lateral plate has not been explored. *Meis1* loss-of-function mutants have hematopoietic, vascular, and ocular defects[35–37] but show no alterations in limb development. Limb skeletal patterning is similarly unaltered in *Meis2* mutants[38,39]. Given that *Meis1* and *Meis2* encode highly similar proteins, both able to inhibit limb distalization and co-expressed during limb development, we reasoned that the lack of phenotype in the single mutants is likely due to redundancy. Here, we describe double *Meis1* and *Meis2* elimination in the limb-forming region and report Meis functions in limb induction and AP and PD patterning.

## Results

**Meis1 and Meis2 are required for limb outgrowth**. To determine the combined role of *Meis1* and *Meis2* in limb development, we conditionally deleted both genes. For this, we generated a new *Meis2*-targeted conditional allele (Supplementary Fig. S1). *Meis1* and *Meis2* deficiency was induced by combining the *Meis2*-targeted conditional allele with either *Meis1* constitutive deletion[35] or a *Meis1*-targeted conditional allele[40]. In both cases, conditional alleles were activated with the *HoxB6CreER* driver[41] induced by 4-hydroxy-tamoxifen (4HT) injection at E8.5 (Fig. 1a). *HoxB6CreER* drives recombination in the posterior 2/3 of forelimbs (FL) and the whole lateral plate posterior to the FL, including the HL forming region (Fig. 1). Using these strategies, disruption of all four alleles resulted in the elimination of all wild-type Meis protein from the recombined regions, as determined by immunofluorescence (Fig. 1c, d). The elimination of *Meis1/2* produced a very strong arrest of limb development, with only vestigial chondrogenic condensations in the FL, despite incomplete recombination there. In the HL, near-complete ($N = 3/9$) or complete ($N = 6/9$) elimination of limb-bud growth and chondrogenic condensations is observed (Fig. 1e, white arrowhead). *Meis1/2* double knockout (from now on, M1KO;M2KO) embryos die around E13.5-E14.5, due to hematopoietic defects.

Cell death and proliferation were evaluated in mutant limbs to test the possibility that elimination of chondrogenic precursors might be the cause of the phenotype (Supplementary Fig. 2).

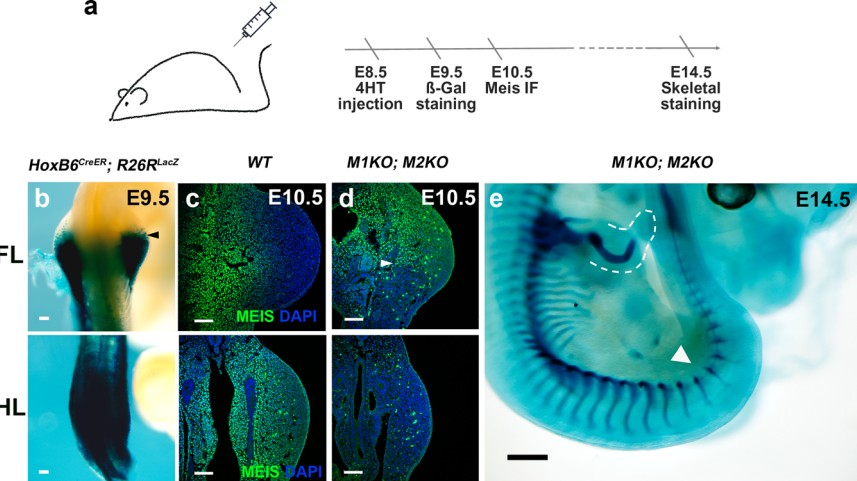

**Fig. 1 Deletion of Meis genes in the limb-forming region. a** Tamoxifen injection and experimental timeline. **b** Whole-mount X-gal staining in E9.5 *HoxB6CreER; R26RlacZ* embryos ($N = 6$ embryos). Scale bar = 100 µm. **c**, **d** Meis immunofluorescence at E10.5 in wild-type and M1KO;M2KO forelimbs (FL) and hindlimbs (HL) ($N = 4$ limbs). Scale bar = 100 µm. **e** M1KO;M2KO limb phenotype ($N = 9$ embryos). Scale bar = 500µm. Arrowheads in **b** and **d** indicate the limits of Cre recombination. In **e**, the dashed line delineates the FL soft tissues and the arrowhead indicates the HL region. Source images are available at Mendeley Data[80] [https://data.mendeley.com/datasets/r774bxyf8d/2].

Mutant FL bud mesoderm showed a clear increase in cell death localized to the central-distal region, whereas the mutant HL bud mesoderm showed a very mild increase (Supplementary Fig. 2a–c, e–g). In FLs, only 2/3 of the limb bud lack Meis activity, and some AER formation and growth are detected. In contrast, HLs are totally devoid of Meis function and do not form a functional AER (Supplementary Fig. 2). The increased cell death thus may contribute to post-induction elimination of prospective limb structures in FLs, however, it is unlikely to account for the complete agenesis of the HL. The limited increase in cell death in the mesoderm of the HL was accompanied by an increase of cell death in the ectoderm (Supplementary Fig. 2g), which is a typical observation in limb buds deficient in mesoderm-to-ectoderm FGF signaling[42]. In regard to cell proliferation, no significant changes were observed in the mesoderm or ectoderm of either FLs or HLs (Supplementary Fig. 2d, h).

These results indicate redundancy between *Meis1* and *Meis2* in limb development and show an early role for Meis TFs in the establishment of limb buds. The early loss of limb buds in M1KO;M2KO embryos precluded further analysis of the effect of complete Meis elimination on skeletal patterning. We, therefore, characterized the skeleton of limbs with at least one intact Meis allele (Fig. 2). As previously reported[35,39], single homozygous deletion of *Meis1* or *Meis2* had no effect on limb skeletal patterning (Fig. 2b, e, g). In contrast, skeletal patterning defects were observed along the PD and AP axes of limbs developing in the absence of three *Meis1/2* alleles (one deleted *Meis1* allele and homozygous deletion of *Meis2*; M1HT;M2KO) (Fig. 2c, f, g).

Skeletal elements proximal to the stylopod-zeugopod boundary were 20–40% reduced along the PD axis in M1HT;M2KO limbs compared to WT limbs, whereas skeletal elements distal to this boundary showed no PD reduction in the mutants (Fig. 2g). Along the AP axis, the posterior zeugopod and autopod elements in M1HT;M2KO limbs were missing or presented an altered anatomy. Alterations included tibial bending (5/10), fibula loss (4/10), and loss or modification of posterior digits (6/10), confirming failure in limb posteriorization (Fig. 2h).

**Molecular analysis identifies the Wnt, Fgf, and Shh pathways as Meis targets.** To characterize the role of Meis transcription factors during limb initiation, we performed an RNA-seq analysis comparing wild-type and M1KO;M2KO HL buds at E10.5 (30 somites). This analysis identified 196 downregulated and 242 upregulated transcripts with an adjusted *p*-value < 0.005 (Fig. 3a and Supplementary Data 1). Downregulated genes included members of the Fgf and Wnt canonical signaling pathways, like *Fgf8* (fold change (FC) −19,1), *Fgf10* (FC −1,5), and *Lef1* (FC −1,4) (Fig. 3b). In addition, the analysis showed changes in genes with important roles in AP and PD patterning (Fig. 3b). M1KO;M2KO limb buds showed downregulation of proximal limb markers and genes involved in proximal limb development, such as *Alx1*(FC −2,9), *Alx3* (FC −1,5), *Shox2* (FC −2,2) and *Tbx15* (FC −1,9)[43–46], and upregulation of distal markers, such as *Tfap2b* (FC 1,9) and *Pknox2* (FC 2,6)[47,48] (Fig. 3b). These changes in PD patterning genes are consistent with the proposed role of Meis in promoting proximal limb fates[31,33].

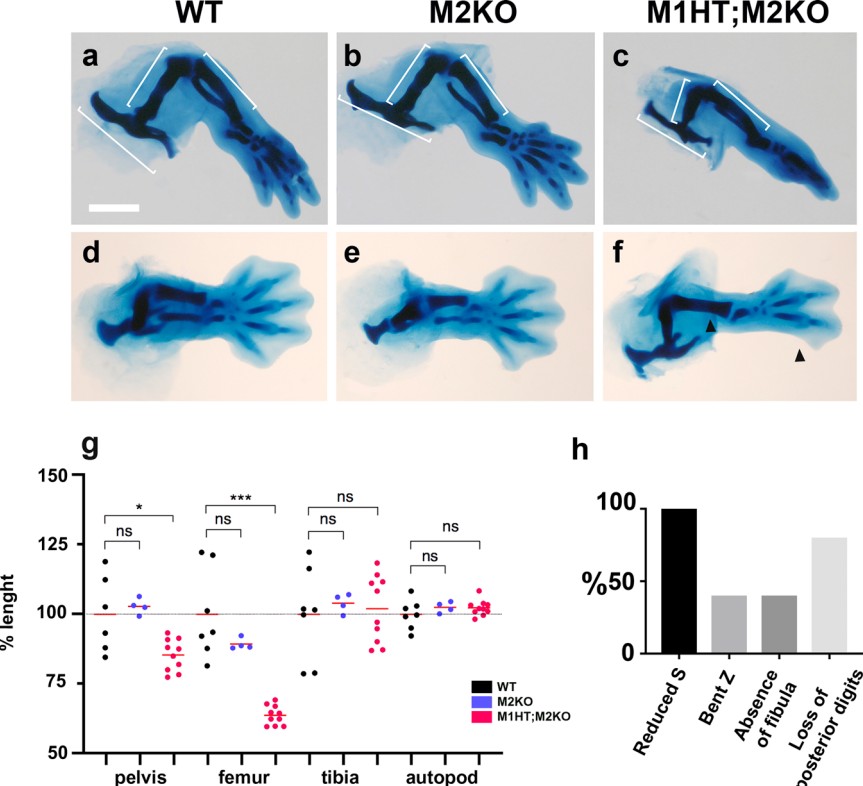

**Fig. 2 Skeletal defects in limb buds lacking three Meis1/2 alleles. a–f** Victoria blue cartilage staining of HLs at E14.5 in WT embryos, with homozygous deletion of *Meis2* (M2KO), and M2KO embryos compound with a *Meis1* knockout allele in heterozygosity (M1HT;M2KO). Scale bar = 1000 μm. **g** Percentage size change in skeletal elements of M2KO and M1HT;M2KO HLs. Limbs analyzed: WT N = 6; M2KO N = 4 and M1HM2KO N = 10. ns, non-significant; *p < 0.05, **p < 0.01, ***p < 0.001. Two-tailed, unpaired Mann–Whitney test. Red bars are mean values. Exact *p*-values from left to right: 0.7619; 0.0420; 0.4121; 0.0001; 0.6485; 0.9623; 0.3152; 0.3148. **h** Percentage occurrence of skeletal alterations in M1HT;M2KO HLs. Arrowheads in **f** indicate the absence of fibula and posterior digits in M1HT;M2KO specimens. N = 10 embryos. Source data are provided as a Source Data file. Source images are available at Mendeley Data[80] [https://data.mendeley.com/datasets/r774bxyf8d/2].

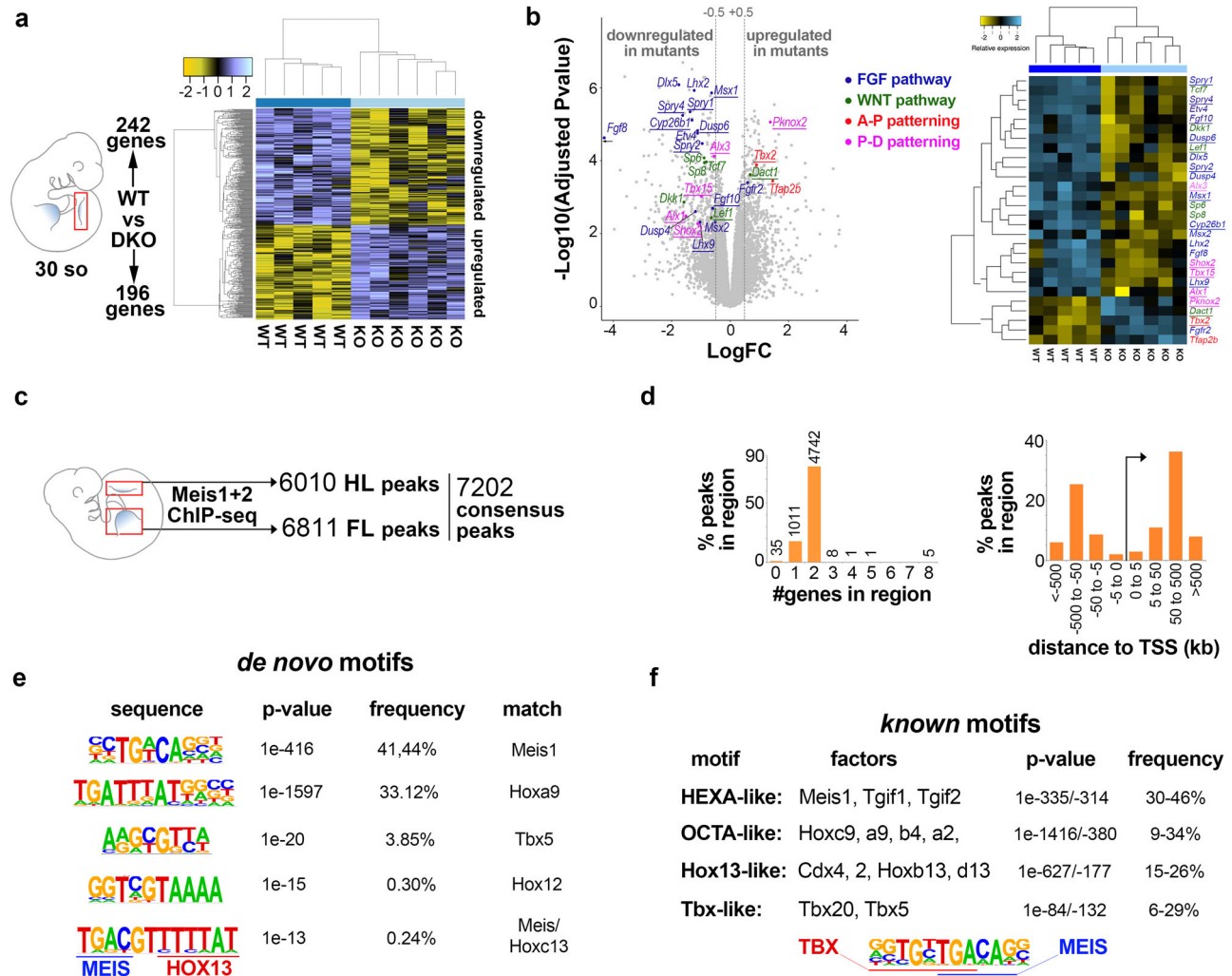

**Fig. 3 Molecular analysis of Meis function in limb buds. a** RNA-seq analysis comparing wild-type and M1KO;M2KO HLs, showing the clustering of WT and mutant samples by expression profile. **b** Volcano plot and heat map with the most relevant genes involved in AP patterning, PD patterning, Wnt and Fgf pathways highlighted in color code. *p*-values are obtained by a Benjamini–Hochberg procedure, which adjusts for multiple comparisons. The underlined highlighted genes show Meis-binding sites in their regulatory regions by ChIP-seq. **c–f** ChIP-seq analysis, showing the number of binding sites identified in E10.5 FLs and HLs (**c**), the distribution of Meis peaks with respect to transcriptional units (**d**) and de novo (**e**), and *known* (**f**) sequence motif identification. FC: fold change. See the "Data availability" section for source data.

To identify genomic Meis-binding sites in limb-bud cells, we next performed a ChIP-seq analysis of FL and HL buds at E10.5. From 6811 peaks detected in FLs and 6010 peaks in HLs (Supplementary Data 2), a total consensus set was identified consisting of 7202 peaks. The consensus list includes 5803 (80%) peaks shared between FLs and HLs, 1008 FL-specific peaks, and 207 HL-specific peaks (Fig. 3c). The binding profile of Meis1/2 in limb buds showed a similar general distribution to that previously described in other embryonic tissues[24,49], characterized by highly conserved sequences and a preference for distal enhancers (Fig. 3d).

Functional enrichment analysis of genes located in the vicinity of the FL-HL shared peaks identified Gene Ontology categories related to limb development as the most significant organ-specific gene sets (Supplementary Fig. 3 and Supplementary Data 3). This observation indicates that Meis binds to genes that are actively transcribed in limb buds and functionally relevant to limb development. Several of the genes close to Meis-binding sites are involved in PD and AP limb patterning and are sensitive to Meis loss in the transcriptomic analysis (Fig. 3b and Supplementary Fig. 4).

A gene-set enrichment analysis against the MSigDB pathway database (http://software.broadinstitute.org/gsea/msigdb/) identified the Hedgehog and Wnt pathways as highly significant gene sets associated with Meis-binding sites (Supplementary Fig. 3). Moreover, although the MSigDB analysis did not detect significant enrichment for Fgf signaling, several core Fgf pathway components were associated with Meis peaks (Supplementary Fig. 5). Significant association with Meis-binding sites was also detected for a set of genes encoding enzymes involved in heparan sulfate and heparin (HS-GAG) biosynthesis, an essential pathway for Fgf signaling (Supplementary Fig. 3). The detection of the Hedgehog pathway in the ChIP-seq analysis correlates with the AP patterning changes identified by transcriptome analysis. The ChIP-seq peaks identifying Meis regulation in Fgf and Wnt pathway genes are shown in Supplementary Fig. 5.

A de novo motif enrichment analysis of Meis-binding regions identified motifs previously reported in the E11.5 trunk ChIP-seq[24], including the hexameric Meis-only-binding sequence and the octameric Hox-Pbx-binding sequence[50] (Fig. 3e). We also found limb-specific motifs not previously identified in the E11.5 trunk. These motifs include binding sequences preferred by the

Hox paralog groups 12 and 13[51], including a putative Meis–Hox13 compound-binding site (Fig. 3e). This observation suggests direct Meis–Hox12/13 interaction in vivo, which correlates with previous biochemical studies showing that group 11-13 Hox proteins cannot directly interact with Pbx but do so through interaction with Meis proteins[50].

In addition, we found a motif related to a known Tbx5-binding site (Fig. 3f). The search for *known* DNA target motifs confirmed the results obtained in the de novo motif search and identified a previously described Tbx5-binding motif in cardiac cells[52], that may be a Tbx–Meis compound motif (Fig. 3f). The detection of Tbx5-binding sequences was highly sensitive to the classification of Meis-binding sites into that co-bound by Tbx5 and those only bound by Meis (Supplementary Fig. 6), further indicating the specificity of the findings.

**Meis is required for limb initiation and extensively co-binds with Tbx and Hox factors in limb buds**. To confirm the RNA-seq data, we next studied the expression of *Fgf8*, which encodes the main AER signal required for limb outgrowth[53,54]. AER-FGF8 mRNA expression was absent ($N = 9/10$) or very weak ($N = 1/10$) in E10.5 M1KO;M2KO hindlimb buds (Fig. 4c, d). Consistently with the Cre recombination pattern, *Fgf8*

expression in the FL was more variable, ranging from complete loss ($N = 3/6$) to reduced expression ($N = 3/6$) (Fig. 4a, b). Activation of *Fgf8* requires the previous activation of *Fgf10* in the limb mesoderm. We found that *Fgf10* mRNA expression was very reduced or absent in M1KO;M2KO early FL buds. The posterior FL bud, where *Meis* genes are inactivated, was found especially affected by this reduction ($N = 8/10$). In HL buds, where *Meis* genes are completely eliminated, *Fgf10* expression was absent or very reduced throughout the buds ($N = 5/6$) (Fig. 4e–h). During limb initiation, *Fgf10* is activated by Tbx5 in FLs[55,56] and by Tbx4 in HLs[57,58]. Expression of the Tbx mRNAs was unaltered in M1KO;M2KO FLs (*Tbx5*) and HLs (*Tbx4*) ($N = 13/13$ and 8/8, respectively) (Fig. 4m–p).

Tbx4/5 activates *Fgf10* by two mechanisms: direct activation of *Fgf10* enhancers and activation of the canonical Wnt signaling pathway effector Lef/Tcf. Both Tbx4/5 and Lef/Tcf then cooperate in *Fgf10* activation[55]. *Lef1* expression was completely absent from the posterior part of mutant FLs buds ($N = 4/4$) and from the whole-mutant HL buds ($N = 4/4$), demonstrating that the Wnt pathway is inactivated in M1KO;M2KO limbs (Fig. 4i–l).

Meis activity is thus involved in the activation of two targets of Tbx4/5 activity essential for limb initiation, *Fgf10* and *Lef1*. These results thus suggest cooperation between Tbx and Meis factors during limb initiation.

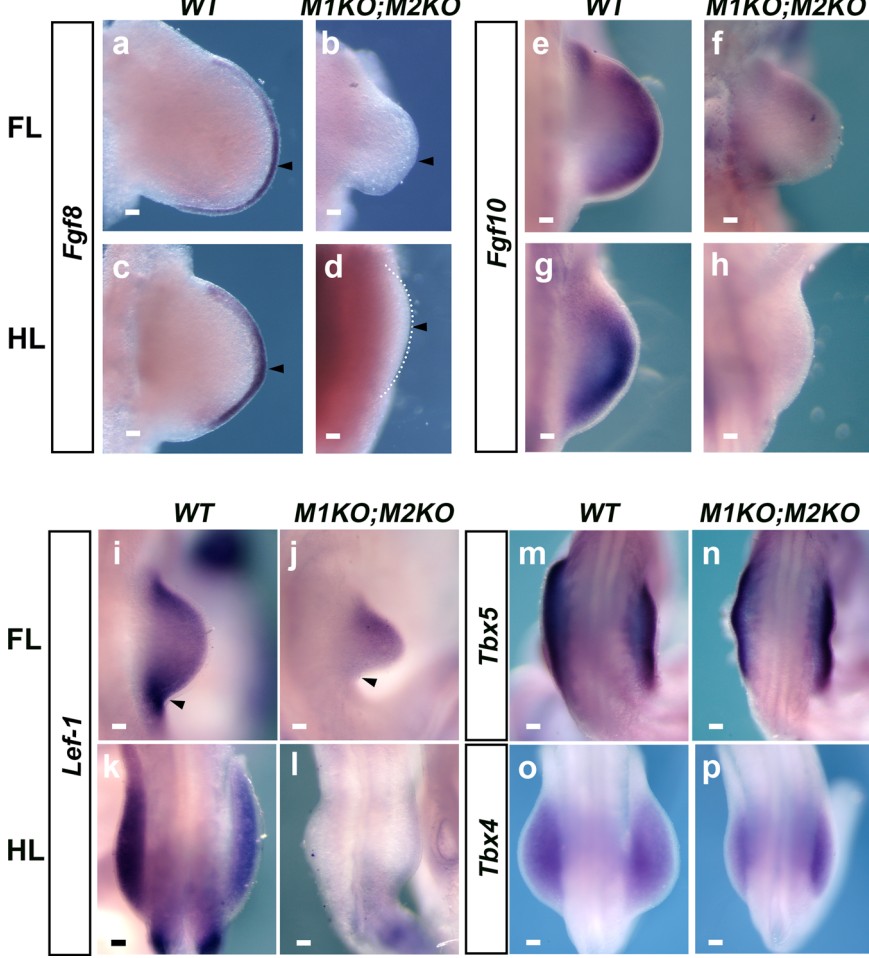

**Fig. 4 Expression of factors required for limb initiation in Meis mutants.** Whole-mount in situ hybridization analysis of the expression of *Fgf8* (**a–d**), *Fgf10* (**e–h**), and *Lef1* (**i–l**) at E10.5 in WT and M1KO;M2KO limbs. For *Fgf8*, $N = 6$ FLs and $N = 10$ HLs. For *Fgf10*, $N = 10$ FLs and $N = 6$ HLs. For *Lef1*, $N = 4$ limbs for both, FLs and HLs. **m**, **n** *Tbx5* expression in E9.5 FLs. **o**, **p** *Tbx4* expression in E10.5 HLs. $N = 13$ limbs for Tbx5 and $N = 8$ limbs for Tbx4. Scale bars = 100μm. Arrowheads in **a–d** indicate the region of *Fgf8* expression in control limb buds and its absence in mutants. Arrowheads in **i**, **j** indicate the expression of *Lef1* in control limb buds and its absence on the posterior part of mutant limb buds. Source images are available at Mendeley Data[80] [https://data.mendeley.com/datasets/r774bxyf8d/2].

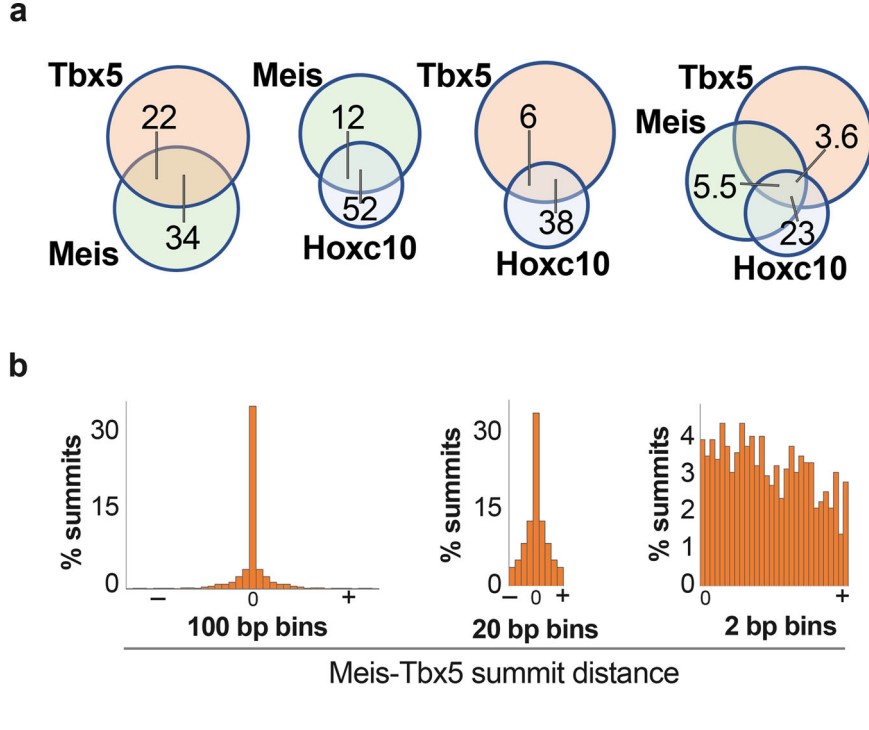

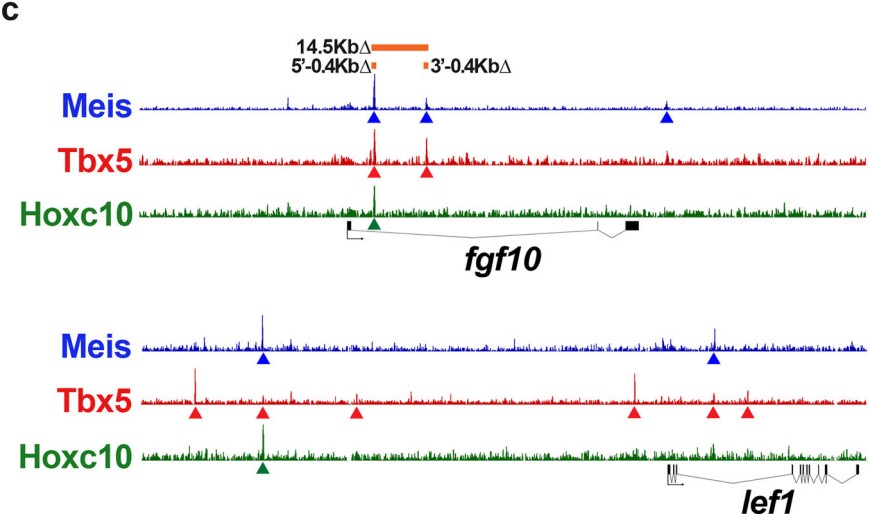

**Fig. 5 Meis, Tbx, and Hox transcription factors co-bind regulatory sequences of genes involved in limb initiation. a** Overlap (%) between ChIP-seq-binding sites for Tbx5, Hoxc10, and Meis. The size of the circles representing each class is proportional to the number of binding sites for each factor (Meis: 6811; Tbx5: 10273; Hoxc10: 1611). **b** Meis–Tbx5 summit distance distribution at different resolutions. **c** Binding sites shared by Meis, Tbx5, and Hoxc10 in the *Fgf10* and *Lef1* loci. See the "Data availability" section for source data.

The presence of Tbx5 motifs in the sequences bound by Meis in ChIP-seq (Fig. 3e, f) suggested co-binding of Tbx and Meis factors to regulatory sequences relevant to limb development. To investigate this possibility, we profiled Meis and Tbx5 genomic-binding sites in E10.5 limb buds by comparing ChIP-seq data from this study with data previously published for Tbx5[59]. Of the 10,267 Tbx5 peaks and 6811 Meis peaks in FLs, 2263 were overlapping peaks (summits at ≤100 bp distance) (Fig. 5a and Supplementary Data 4). The distribution of distances between Meis and Tbx5 summits was non-random, with a sharp increase in the frequency of summits at distances below 100 bp (Fig. 5b). The preference for shorter distances was maintained among summit pairs closer than 100 bp, with the highest enrichment at

distances <30 bp (Fig. 5b). Nonetheless, there was no evidence for a preferred specific distance between Meis and Tbx5 summits (Fig. 5b), suggesting the absence of predominant specific complexes between these factors that would constrain the distance between binding sites.

Given the predominance of Hox-binding motifs in Meis-bound genomic sequences[24] and the evidence of cooperation between Hoxc10 and Tbx factors in limb buds[60], we investigated the coincidence between Meis, Tbx, and Hoxc10-binding profiles (Fig. 5a and Supplementary Data 4). Although Hoxc10 is expressed in HLs and not in FLs, a strong overlap was found between Hoxc10 and Meis binding in the FL (Supplementary Data 4). Furthermore, Hoxc10-binding sites in HLs overlapped

more extensively with FL Tbx5-binding sites than with HL Tbx4 sites[60]. These results suggest that most Hoxc10-binding sites are general limb-bud Hox target sequences and are not HL-specific. The superior overlap of Hoxc10 with Tbx factors in the FL[60] probably reflects the higher quality of the FL ChIP-seq experiments for these factors. The strong overlap between Hoxc10 and Meis-binding sites (>50% of Hoxc10 sites) is consistent with the established role of Meis as a preferential cofactor for Hox proteins[24] and was more extensive than the overlap between Hoxc10 and Tbx5-binding sites. Binding sites common to Tbx5, Hoxc10, and Meis appeared at a higher frequency than predicted by the pairwise interactions (Fig. 5a). General Meis–Tbx5 overlap affected 34% of Meis peaks and 22% of Tbx5 peaks (Fig. 5a). In contrast, within peaks common to Tbx5 and Hoxc10, Meis–Tbx5 overlap increased to 74% of Meis peaks, while within the Meis–Hoxc10 common peaks, Meis–Tbx5 overlap increased to 44% of Tbx5 peaks (Fig. 5a). These observations suggest that the previously reported Tbx5–Hox interaction[60] also involves Meis factors.

**Regulatory regions bound by Meis, Tbx, and Hox factors are required for *Fgf10* expression in the early limb bud.** Interestingly, Meis–Tbx5 and Meis–Tbx5–Hoxc10 shared binding sites were detected in the vicinity of *Fgf10* and *Lef1*, two of the putative common direct targets of Meis and Tbx essential for limb induction (Fig. 5c). To study the function of the putative regulatory regions identified by the binding of Meis, Tbx5, and Hoxc10 to the *Fgf10* locus, we performed CRISPR/Cas9-mediated targeted deletions in mouse zygotes. Two independent 0.4 kb deletions located in intron 1 were generated. The more 5′ deleted region (5′−0.4Δ) contains Meis, Tbx5 and Hoxc10-binding sites, whereas the more 3′ deleted region (3′−0.4Δ) contains only Meis and Tbx5-binding sites. We also generated a 14.5 kb deletion (14.5Δ) that includes both 0.4 kb deletions and the region between them (Fig. 5c). Homozygosity for either 5′−0.4Δ or 14.5Δ resulted in lower expression of *Fgf10* mRNA in E9.5 FL buds ($N = 12/12$ and 8/8, respectively) (Fig. 6a, solid arrowheads). Homozygosity for 3′−0.4Δ also resulted in lower *Fgf10* expression in the posterior FL bud region and complete elimination of *Fgf10* expression in the anterior FL bud region ($N = 10/10$) (Fig. 6a, empty arrowhead). Downregulation of *Fgf10* expression in FLs was confirmed by qPCR (Fig. 6b).

In 5′−0.4Δ and 3′−0.4Δ, the lower expression is specifically detected in limb buds and does not affect other embryonic regions, such as the eye or tail expression domains (Fig. 6a, asterisks). In 14.5Δ embryos, in contrast, these regions might also show a reduction in *Fgf10* expression. The observed reduction in expression was transient and by E10.5 *Fgf10* expression recovered in embryos homozygous for the different deletions and limb outgrowth progressed normally (Fig. 6a).

While these results showed that the characterized enhancers were not essential for limb development, they also suggested that their absence would represent a background sensitized to the loss of transcription factors involved in their regulation. To study this possibility, we generated stable mouse lines for the 5′−0.4Δ and 14.5Δ deletions and studied their synergistic interactions with Meis mutations. For this purpose, we generated M1HT;M2KO; 5′−0.4Δ and M1HT;M2KO; 14.5Δ embryos and compared their phenotypes to M1HT;M2KO embryos (Fig. 6c–f). The combined Meis and FGF-enhancer mutants displayed stronger defects than Meis mutants alone, demonstrating a positive functional interaction. Increased incidences of fibula and digit loss were found in the compound mutants (Fig. 6e). In order to quantify these defects, a score was used that assigns each missing element 1 point (i.e., fibula and two digits lost = 3 points). As shown in the

graph, both 5′−0.4Δ and 14.5Δ deletions significantly worsen the limb phenotypes of M1HT;M2KO animals, despite they do not produce any defect in isolation (Fig. 6f). These results indicate that in a Meis loss-of-function background, the identified *Fgf10* regulatory regions become functionally relevant.

Regarding the length of skeletal elements, unexpectedly, the introduction of the *Fgf10* regulatory region deletions resulted in the rescue of the stylopod length, normally reduced in M1HT; M2KO animals. The stylopod length is increased around 25% by either the 5′−0.4Δ or the 14.5Δ deletions (Fig. 6c, d). These results suggest that proximal defects due to Meis impairment can be rescued by reducing distal Fgf signaling, an observation that aligns with the idea of opposing proximal and distal pathways in PD limb patterning[31,33].

These results identify roles for the characterized sequences in regulating early *Fgf10* expression and show complex interactions between them. Meis, Tbx, and Hox transcription factors thus converge in the regulation of the Wnt and Fgf pathways during limb initiation.

**Meis activity is essential for limb-bud AP prepatterning.** We next investigated the molecular basis of the defects in AP patterning. *Shh* mRNA expression in M1HT;M2KO FLs and HLs was either strongly reduced ($N = 4/4$ FLs and 1/3 HLs), absent ($N = 2/4$ HLs) or misplaced ($N = 1/4$ HLs) (Fig. 7a), which is sufficient to explain the observed AP phenotypes. Correlating with the absence of Shh, the direct Shh activation target *Ptch1*[61] was downregulated in the mutants (4/4). In addition, expression of the Shh-repressed factor *Cdon*, normally restricted to anterior limb-bud regions[62], extended toward the posterior limb region (4/4) (Fig. 7a).

We next investigated upstream *Shh* regulators. These include Gli3 and Hand2, whose antagonism establishes an AP prepatterning required for *Shh* activation. In M1KO;M2KO mutant FLs, *Hand2* expression was lost or very reduced (6/6), while *Gli3* expression, normally confined to the anterior part of the limb bud, extended posteriorly (2/2) (Fig. 7b, c). Strikingly, *Hand2* expression in mutant HLs was not repressed but instead extended across the whole HL bud AP axis (4/6). This was accompanied by the presence of low, homogeneous *Gli3* expression across the HL bud AP axis (2/2) (Fig. 7b, c). Thus, although AP polarization was altered in both FLs and HLs, the specific alterations in each of them were different.

Different phases of Hox expression have been implicated in the generation of the limb AP prepattern and subsequent *Shh* activation. *Hoxd10-13* genes are required for *Hand2* activation and cooperate with Hand2 in direct *Shh* activation[18,19,22,23].

Although in late-stage wild-type limb buds *Hoxd13* is confined distally, at early stages it is found in the posterior limb bud, in agreement with its role in regulating *Hand2* and *Shh* expression. In M1KO;M2KO FLs, *Hoxd13* expression was either absent (4/8) or strongly reduced (4/8) (Fig. 7d). In contrast, M1KO;M2KO HLs showed only very subtle downregulation (8/8). These observations highlight the different mechanisms underlying Meis function and AP limb prepatterning in FLs and HLs. Before *Hoxd10-13* is activated in the limb bud, *Hox9* paralog group activity is required in the lateral plate to establish the limb field AP polarity[13]. At the limb-bud initiation phase, *Hoxd9* expression extends posteriorly from the limb bud into the lateral plate. This extended expression is not observed in the anterior part of the limb bud, where there is a sharp border of *Hoxd9* expression. In M1KO;M2KO mutants, the extended posterior expression was not observed in FLs (3/4) (Fig. 7e). Instead, the posterior region of limb buds displayed a sharp border of *Hoxd9* expression, producing a symmetrical expression domain along the AP axis.

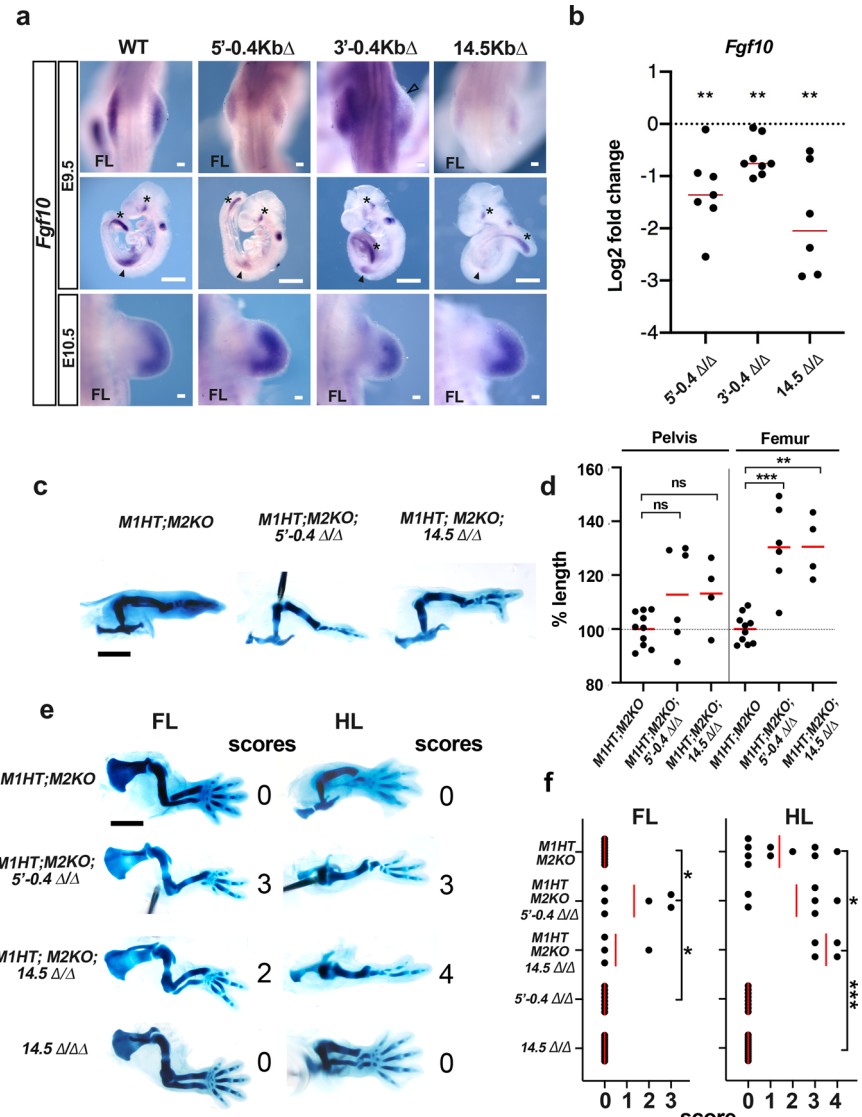

**Fig. 6 Regulatory elements bound by Meis, Tbx, and Hox factors control Fgf10 expression at limb initiation. a** *Fgf10* expression at E9.5 and E10.5 in embryos with the CRISPR/Cas9-mediated-targeted *Meis* deletions indicated in **c**. Solid arrowheads indicate *Fgf10* expression domain in FLs. Asterisks indicate non-limb *Fgf10* expression domains. Empty arrowhead indicates the specific absence of *Fgf10* expression in the anterior limb bud. Limbs analyzed: N = 12 for 5′−0.4Δ, N = 10 for 3′−0.4Δ, and N = 8 for 14.5Δ. Scale bars = 100 µm in FLs and 1000 µm in whole embryos. **b** Quantitative RT-PCR showing a reduction of *Fgf10* transcripts in E9.5 FL buds of embryos with the indicated regulatory region deletions. Limbs analyzed: For 5′−0.4Δ, N = 7; For 3′ −0.4Δ, N = 8. For 14.5Δ, N = 6. Two-tailed, one-sample *t*-test. Exact *p*-values from left to right: 0.0037; 0.0013; 0.0081 respectively. Red bars represent median values. **c** Examples of limb skeletons of genetic combinations between Meis mutants and *Fgf10* regulatory region deletions. Scale bar = 1000 µm. **d** Comparison of pelvis and femur length in genetic combinations of Meis mutants and *Fgf10* regulatory region deletions. Two-tailed, unpaired Mann–Whitney test. Limbs analyzed: M1HTM2KO N = 10; M1HTM2KO;5′−0.4Δ, N = 6; M1HTM2KO;14.5Δ, N = 4. Exact *p*-values from left to right: 0.3132; 0.0759; 0.0019; 0.0020. Red bars represent mean values. **e** Examples of limb skeletons of genetic combinations between Meis mutants and *Fgf10* regulatory region deletions, showing the limb AP axis. Scale bar = 1000 µm. **f** Scoring of skeletal element loss in the genetic combinations of Meis mutants and *Fgf10* regulatory region deletions shown in **c**. Loss of each skeletal element (fibula or digit) scores 1. Embryos analyzed: M1HTM2KO N = 10; M1HTM2KO;5′−0.4Δ, N = 6; M1HTM2KO;14.5Δ, N = 4; 14.5Δ N = 12; 5′−0.4Δ, N = 8. Two-tailed, unpaired Mann–Whitney test. Exact *p*-values from top to down and left to right: 0.0357; 0.0245; 0.0360; 0.0005. Red bars represent mean values. ns, non-significant; *$p < 0.05$, **$p < 0.01$, ***$p < 0.001$. Statistics were not adjusted for multiple comparisons. Source data are provided as a Source Data file. Source images are available at Mendeley Data[80] [https://data.mendeley.com/datasets/r774bxyf8d/2].

This can also be observed by Hox9 immunofluorescence (Fig. 7e, white arrowhead; N = 3/3).

These results show that Meis is required for *Shh* activation through its role in regulating genes involved in AP limb prepatterning, including *Gli3*, *Hand2*, and 5′ *Hoxd* genes. The ChIP-seq results support a direct regulatory role for Meis in some of these interactions. Meis-binding sites were detected in the vicinity of *Hand2*, *Ptch1*, and the *HoxD* complex (Supplementary Fig. 4). A very prominent Meis peak is detected in *Hoxd13*, which represents the only reported instance of Meis-binding 5′ to the *Hox9* paralog position in any Hox complex. This interaction is specific to limb buds, since it was not detected in E11.5 whole-trunk tissues[24]; moreover, it is stronger in FLs, which correlates with the exclusive dependence of *Hoxd13* expression on Meis in FLs (Fig. 7d).

Another hint pointing to a relevant Meis–Hoxa13–Hand2 interaction is that the described *Hand2* limb enhancer mm1689[63]

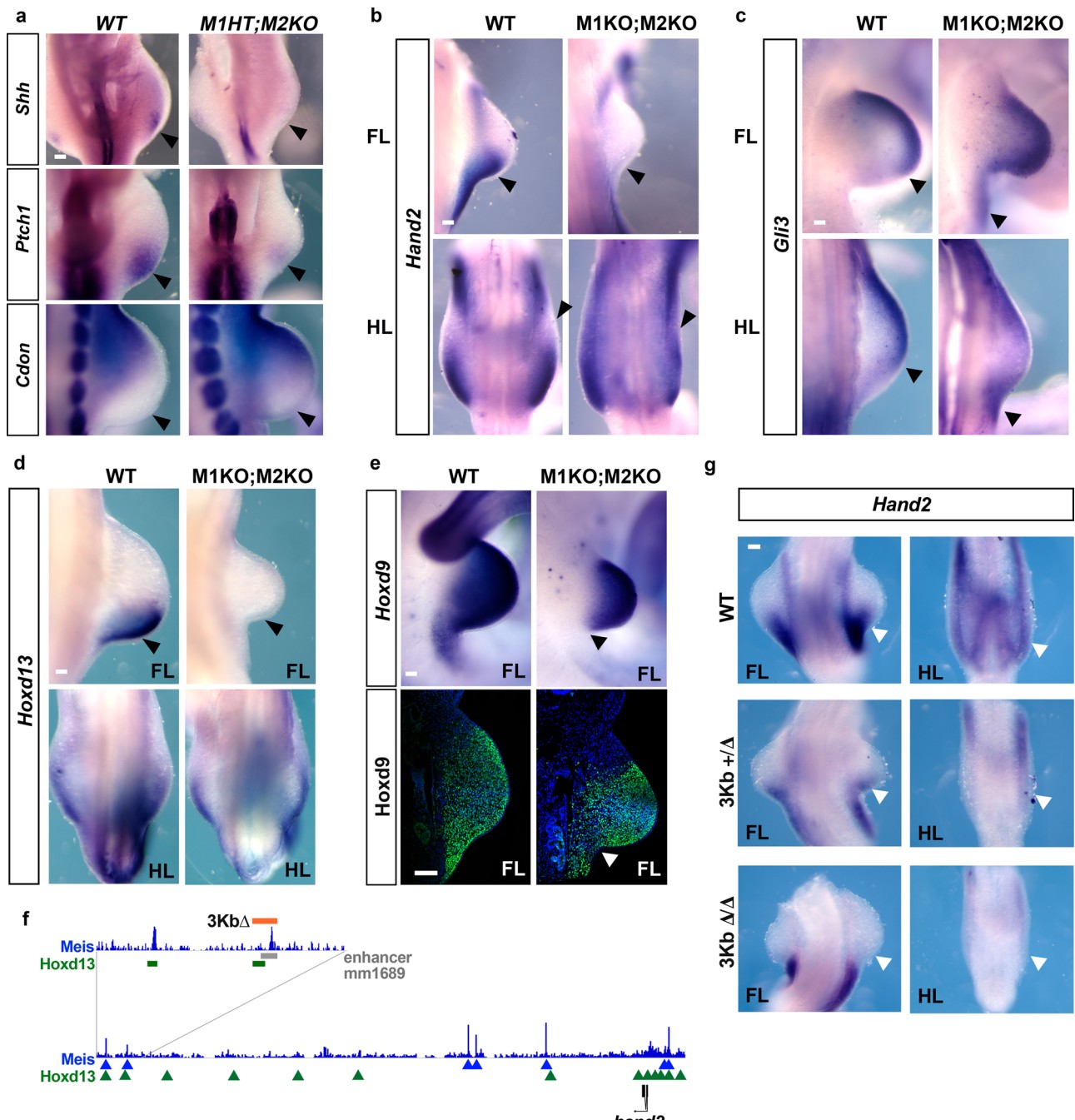

**Fig. 7 Meis function in limb-bud AP prepatterning. a** Whole-mount in situ hybridization analysis of *Shh* and its targets *Cdon* and *Patched1* (*Ptch1*) in WT and M1HT;M2KO HLs at E10. Limbs analyzed: *N* = 4, *N* = 4, and *N* = 2, respectively. Whole-mount in situ hybridization analysis of the expression of *Hand2* (**b**) (*N* = 6 FLs and *N* = 6 HLs), *Gli3* (**c**) (*N* = 2 FLs and *N* = 2 HLs), *Hoxd13* (**d**) (*N* = 8 FLs and *N* = 8 HLs) and *Hoxd9* (**e**, upper panels) (*N* = 4 FLs) at E10.5 in WT and M1KO;M2KO FLs and HLs. Scale bars = 100 µm. (**e** lower panels) HOXD9 immunofluorescence in E10.5 FLs (*N* = 3 FLs). Scale bar = 100 µm. **f** Map of the Hand2 locus with its regulatory regions and ChIP-seq signal for Meis and HOXD13[64]. The position of the mm1689 enhancer (gray bar), the associated Meis and HOXD13-binding sites (blue peaks and green bar, respectively), and the 3 kb region deleted (orange bar) are shown in a zoomed region. **g** Whole-mount in situ hybridization analysis of *Hand2* at E10 in embryos carrying the 3 kb regulatory region deletion (*N* = 4 homozygous FLs and HLs and *N* = 4 heterozygous FLs and HLs). Arrowheads point to protein or gene expression downregulation. Source images are available at Mendeley Data[80] [https://data.mendeley.com/datasets/r774bxyf8d/2].

overlaps with Meis and Hoxd13[64]-binding sites (Fig. 6f). To study the functionality of this enhancer during limb AP patterning, we used the CRISPR–Cas9 system to delete a 3-kb region comprising the *Hand2* enhancer, Meis, and Hoxd13-binding sites (Fig. 7f). Heterozygosity for this deletion resulted in limb-specific *Hand2* downregulation (*N* = 2/4 in both FLs and HLs), whereas

homozygosity led to limb-specific complete absence of *Hand2* expression (*N* = 4/4 in both FLs and HLs) (Fig. 7g).

In contrast to these observations, Meis binds neither the *Shh* locus, nor the ZPA regulatory sequence (ZRS), essential for *Shh* activation (Supplementary Data 1). This is unexpected, given that the ZRS is bound by Pbx and Hox factors[19]. Our results thus

indicate that Meis contributes to forelimb AP patterning through its involvement in establishing *Hand2* and *Hox* gene expression.

## Discussion

Here, we used a conditional targeting approach to study the effect of complete *Meis* loss-of-function on early limb bud development. *Meis1* and *Meis2* double knockout embryos are limbless, with rudimentary or absent appendicular chondrogenic condensations. Consistent with these phenotypes, *Fgf10* and *Lef1* are severely downregulated or lost. Similar limbless phenotypes were reported for *Fgf10* knockouts[6,8] and *Lef-1/Tcf* double knockouts[65], although the phenotypes were slightly less severe in HLs. This possibly reflects the action of Meis upstream of both *Fgf10* and *Lef1*. In contrast, in FLs the phenotypes of *Lef-1/Tcf* mutant are more severe than those of Meis mutants, very likely reflecting the incomplete recombination driven by *HoxB6CreER* in FLs. The strong phenotype observed suggests that no other related transcription factor can compensate Meis function in limb development, despite the overactivation of related factors, like *Pknox2*.

The limb-bud ChIP-seq analysis identified previously known Meis DNA-binding motifs, but also detected Tbx-binding motifs and a putative Meis–Tbx compound motif, suggesting that these transcription factors cooperate in limb development. Interestingly, enrichment of Meis motifs in Tbx5 genomic-binding sites was reported in a ChIP-seq analysis of Tbx5 during cardiac differentiation[66]. In limb buds, Tbx5 genomic-binding sites[59] overlap with ~1/3 of Meis-binding sites, suggesting strong cooperation between these two transcription factor families. We found binding sites shared by Meis and Tbx5 in the enhancers of *Fgf10* and *Lef1*, both of which are critical genes for limb induction and depend on Meis activity for their expression. Confirming the enhancer function of Tbx5/Meis-bound elements in *Fgf10*, deletion of these sites impaired *Fgf10* expression in limb buds at the limb initiation step. Nonetheless, the impairment observed was incomplete and transient, not producing any further alterations of limb patterning. The combination of enhancer deletion with Meis deletion, however, did enhance the skeletal defects, demonstrating the functional relevance of these regulatory sequences. These observations suggest that additional Meis-target sequences should be involved and that compensatory cis-mechanisms restore normal levels of *Fgf10* expression in the isolated enhancer deletions. One possibility is that the feedback activation of *Fgf10* by AER-Fgfs depends on different regulatory elements than those used by Meis/Tbx to initially activate it.

Interestingly, not only are Hox proteins not excluded from the Meis–Tbx interaction, but Hox participation seems to promote it, since Meis–Tbx interactions were much more frequent within Hoxc10-binding sites. These results suggest that Tbx, Hox, and Meis cooperate in organ patterning. This cooperation may be relevant to other organs, such as the heart, where Meis1 and Meis2, anterior Hox genes, and Tbx5 are co-expressed in the posterior heart field[67].

Meis genes are essential not only for limb initiation but also for establishing AP prepatterning and Shh induction. *Hand2* and *Hoxd13*, both required for *Shh* transcriptional activity, were not expressed in FLs of Meis double KO embryos, providing a sufficient explanation for failed *Shh* activation. While Meis double KO HLs also fail to activate Shh and show alterations in the expression of prepatterning genes, such as *Hand2* and *Gli3*, the specific regulatory interactions here remain to be further studied. Meis-binding sites were found in the *Hand2* and *Hoxd13* loci, suggesting direct regulation of both. Meis binding to *Hoxd13* is an exception to the profile of Meis binding to Hox complexes. In a previous ChIP-seq analysis of E11.5 embryonic trunk, dense Meis

binding was detected in Hox complex regions containing paralogs 1–9, but not to those containing paralogs 10 to 13[24]. We found the same result in limb buds for all complexes except for *HoxD*, where we detected a very high scoring peak in *Hoxd13*. This Meis-binding site was also detected in HLs, but with a much lower score. Interestingly, while *Hoxd13* expression was completely missing in mutant FLs, it was unchanged in mutant HLs. These results indicate a strong correlation between Meis binding and posterior limb-bud *Hoxd13* expression. Interestingly, Hoxd13 binds to *Meis1* and *Meis2* loci in E11.5 limbs, suggesting a reciprocal regulation[15]. Interaction of Meis with Hox paralog group 13 is expected not to be cooperative but rather antagonic during limb PD patterning[31,68]; however, during AP patterning, 5′ *HoxD* and Meis gene expression show considerable overlap. Interestingly, Meis and paralog group 13 proteins interact directly, suggesting that they could cooperate in regulating downstream targets[50]. The enrichment of Meis genomic-binding sites for compound Meis–Hox13 DNA motifs, further suggests cooperation between these transcription factors in limb development. A plausible scenario is that Meis activates *Hoxd13* and then cooperates with Hoxd13 protein in regulating downstream targets during early AP patterning. This idea is supported by the extensive co-binding of these factors identified by Hoxd13 ChIP-seq in E11.5 FLs[64] and by the Meis ChIP-seq presented here in E10.5 FLs. In support of this view, deletion of a Hand2 enhancer that contains Hoxd13 and Meis-binding sites, specifically abolishes Hand2 limb-bud expression. While this cooperation may operate at the early stages of AP patterning, it would not occur at later stages, when Meis is excluded from the distal limb. While here we studied *Hoxd13* in this context, the single elimination of *Hoxd13* does not affect *Shh* activation, which is only affected by the removal of genes *Hoxd10* to *Hoxd13*[22]. The proposed role for Hoxd13 is therefore likely redundant with other HoxD 5′ genes.

A restriction of the AP patterning role of Meis to the limb initiation phase is supported by parallel experiments using Cre drivers that eliminate Meis function after limb induction[31]. These experiments showed a very strong disruption of PD development in *Meis1/2* double KO limbs but no alteration in AP patterning[31].

The PD and AP phenotypes of *Meis1/2* double KO limbs are consistent with those produced by mutations of the Meis partners *Pbx1* and *Pbx2*[19]. Complete elimination of *Pbx1/2* leads to limb agenesis, and elimination of three of the four *Pbx* alleles results in the reduction of proximal elements and absence of *Shh* expression. While cooperation between Meis and Pbx would be the most straightforward interpretation of these results, Pbx factors were found to directly regulate *Shh* transcription[19], whereas here we found no direct interaction of Meis proteins with the *Shh* locus. Thus, while Meis and Pbx might cooperate during limb induction and PD patterning, they seem to have different roles during limb AP patterning.

Our results identify roles for Meis transcription factors in limb development and identify their involvement in pathways essential for limb development (Fig. 8). During limb initiation, Meis factors are essential to induce *Fgf10* and *Lef1* expression (Fig. 8a). This function is similar to that previously described for Tbx factors. Meis, Tbx, and Hox factors co-bind to several regulatory elements of these limb initiation genes. Fgf10 regulatory loop with AER-Fgfs ensures correct limb initiation. This mechanism is shared between FL and HL. Concerning AP patterning, Meis is required in the FL for *Hoxd9* and *Hoxd13* expression in the posterior region of the limb bud (Fig. 8b). The restricted expression of 5′ *HoxD* factors establishes the limb AP prepatterning through *Hand2* activation. Meis also binds directly *Hand2* enhancers essential to regulate its expression. Meis is thus directly or indirectly involved in several steps of limb AP prepatterning essential for *Shh* activation. This coordinated

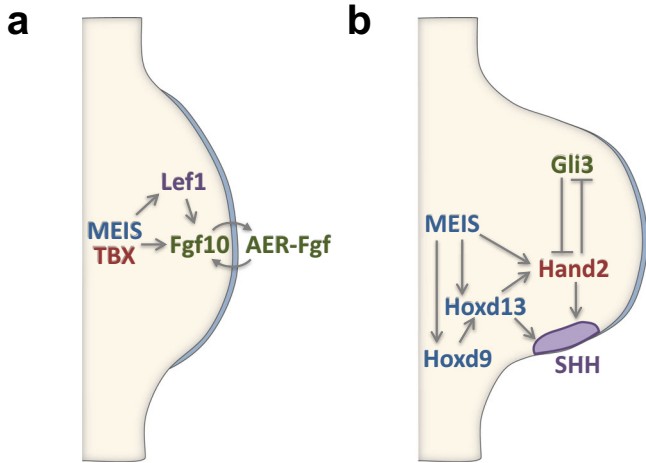

**Fig. 8 Proposed role of Meis in limb initiation and AP prepatterning.**
During limb initiation (**a**), Meis is required in parallel to Tbx factors for activation of the Wnt target *Lef1* and *Fgf10*, both essential for limb-bud formation and progression. This function is similar in FLs and HLs. During AP patterning in the FL (**b**), Meis is required directly or indirectly for the activation of various prepatterning genes, like *Hoxd9*, *Hoxd13*, and *Hand2*, all involved in Shh activation.

regulation leads to *Shh* posterior activation and reciprocal Gli3 expression, establishing AP limb patterning (Fig. 8b). While the outcome of Meis elimination is similar in the HL—failure to activate Shh—the specific regulatory interaction in the HL is different and will require further study.

## Methods

**Mice**. Experiments were performed using mice (*Mus musculus*). Mice were handled in accordance with the CNIC Ethics Committee, Spanish laws, and the EU Directive 2010/63/EU for the use of animals in research. All mouse experiments were approved by the CNIC and Universidad Autónoma de Madrid Committees for "Ética y Bienestar Animal" and the area of "Protección Animal" of the Community of Madrid. The targeting vector used to introduce loxP sites flanking *Meis2* exon 3 was generated by GeneBridges following these steps: (1) insertion of a loxP-flanked neoR cassette into BAC bQM241K9 by Red/ET recombination; (2) subcloning of a 17 kb BAC fragment into minimal vector pMV and subsequent Cre recombination to obtain a single loxP site; (3) insertion of a loxP-FRT-PGK-gb2-neo-FRT cassette into a second location in the sub-cloned DNA. The targeting vector was SalI-released from the backbone and electroporated into ES cells. ES cell clones were screened by Southern blot. Chimeras were generated by ES cell aggregation. The neomycin resistance cassette was eliminated with a Flpe deleter strain[69]. Mice were subsequently genotyped by PCR. All primer sequences for genotyping, qPCR, and CRISPR/Cas deletions are shown in Supplementary data 5.

*HoxB6CreER* mice were as previously described[41]. *Meis1flox* mice (*Meis1tm2Ngc*)[70] were generated by Kenneth Humphries (Terry Fox Laboratory, Vancouver) and kindly provided by Dr. Hesham Sadek (UTSW, Dallas).

4-hydroxy-tamoxifen (4HT) (Sigma) was dissolved in corn oil (5 mg/ml) and injected intraperitoneally (0.8 mg single dose) at E8.5.

**In situ hybridization**. Whole-mount in situ hybridization was performed according to standard protocols[71]. Samples were fixed in 4%PFA, dehydrated in methanol, rehydrated, then a bleaching step (30 min with 6% $H_2O_2$) before proteinase K (PK) treatment. Samples were then hybridized with the probe overnight. After SSC-CHAPs and TBST wash samples were developed with BM purple (Roche).

**Skeletal preparation**. Victoria Blue staining was performed following standard procedures[72]. Embryos were eviscerated and fixed in 10% formaldehyde overnight and then washed in acid alcohol (3% HCl in 70% ethanol) several times. Embryos were stained for 3 h with 0.5% w/v Victoria Blue (Sigma) in acid alcohol and after staining embryos were washed in acid alcohol until the embryos were white, then they were washed in 70% ethanol and 95% ethanol. Finally, embryos were clarified with increasing concentrations of methyl salicylate in ethanol (30%, 50%) and stored in 100% methyl salicylate.

Measurements of long bones were done with ImageJ and individually normalized by each embryo's crown-rump length. Images were taken by using Nikon microscopy camera acquisition software.

**RNA-seq**. E10 hindlimbs were dissected from wild-type embryos (5 samples) and M1KO;M2KO embryos (6 samples). Each sample included both HLs of an individual embryo. RNA was isolated from tissue samples using the RNeasy Mini Kit (QIAGEN). Samples were sequenced using Illumina HiSeq technology. An amplification step (NuGen Ovation) was included in the sample preparation protocol. For the analysis, reads were first preprocessed with Cutadapt 1.3 to eliminate Illumina adaptor remains, and to discard reads that were shorter than 30 bp, using FastQC 0.8.0 to assess read quality after each preprocessing step. The resulting reads were then aligned against a mouse transcriptome reference (GRCm38 assembly, release 70) with RSEM 1.2.3 to estimate gene-level expression levels. Finally, estimated count matrices were used as input in a pipeline for differential expression analysis that included TMM (contained in R package Limma), ComBat (contained in R package SVA 3.20.0 and Limma 3.28.21) as main components. The analysis was restricted to a selection of 13,720 genes that were expressed at a minimal level of 1 CPM in at least 4 samples. Changes in gene expression were considered significant if associated with Benjamini and Hochberg adjusted p-value < 0.05.

**Statistical analyses**. They are detailed in Figure legends. In all cases, individual measurements were taken from distinct samples. Prism 9 software was used for the analyses.

**Immunofluorescence**. Standard procedures were used for immunofluorescence with antigen retrieval step. Anti-Meisa antibody was produced in the lab, recognizing C-terminal short isoform of Meis1 and Meis2[73] (dilution 1/800) and anti-HoxD9 (H-342 Santa Cruz Biotechnology) (dilution 1/100). Images were taken by using Leica microscopy camera acquisition software.

**β-gal reporter analysis**. *HoxB6CreER* transgenic embryos were fixed for 30 min, rinsed in PBS, and incubated in the presence of X-gal as described[74].

**CRISPR/Cas9 deletions**. CRISPR/Cas9 gene editing was performed in the pronuclei of B6CBAF1xB6 zygotes[75] using two single-guide RNAs (sgRNAs) flanking the region of interest. See Supplementary Data 5 for primers. We used the Alt-R® S. p. Cas9 Nuclease V3 from Integrated DNA Technologies. Microinjections were performed by the Transgenesis Unit at CNIC. Some embryos were allowed to proceed to birth to generate mouse lines. Others were collected for transient transgenic analysis at E9.5 and E10.5.

**Cell death and proliferation assay**. Rabbit polyclonal anti-phospho-histone 3 antibody (06-570 Sigma-Aldrich) (dilution 1/200) and TdT-mediated biotin-dUTP nick end labeling (TUNEL) detection kit[76] were used according to the manufacturer's instructions. Quantification was performed manually counting TUNEL or pH3 positive cells and normalizing with DAPI positive nuclei (ITCN plugin in ImageJ) in each section. Five WT FLs, 4WT FL, and 6 M1KO;M2KO limbs were considered and 3 sections of each limb were quantified. Statistical comparisons were performed using a Mann–Whitney test by using Prism 9 software.

**qPCR**. Both FLs of E9.5 embryos containing the different deletions and somite-staged corresponding WT FLs were dissected and frozen individually. RNA purification was done by Qiagen miRNeasy Mini Kit and retrotranscribed by Invitrogen SuperScript™ II Reverse Transcriptase. qPCR was then performed using Sybr Green PCR Master Mix and run in an ABI Prism 7000 from Applied Biosystems. Statistical comparisons were performed using a one-sample t-test. See Supplementary data 5 for primers.

**ChIP-seq**. Around 500 E10.5 (36so) wild-type FLs and 500 E10.5 wild-type HLs were fixed according to standard procedures for ChIP experiments. For the IP, we used ChIP-IT High Sensitivity Columns (Active Motif). Two anti-Meis antibodies were used simultaneously, one recognizing Meis1a and Meis2a,c and the other recognizing all Meis1 isoforms[73]. ChIP DNA was sequenced with an Illumina HiSeq 2500 system by NDeep sequencing at the CNIC Genomics Unit. Single-end reads were mapped with BWA 0.7.15 software against version mm9 of the mouse genome. The alignments were then used for peak calling by MACS 2.1.1, using p = 0.0005 as the cutoff for detection, except for the FL-HL common peaks, for which the cutoff p-value was 0.0005 for the compound occurrence of the same peak in both data sets. Detection of peak-to-gene association and functional association of the identified gene sets were performed with GREAT 3.0.0[77]. A basal regulatory domain was assigned to each gene of extending from 5000 bp upstream of the transcriptional start site to 1000 bp downstream. The regulatory domain is extended in both directions to the nearest gene's basal regulatory domain but no more than 1000 kb from the TSS in each direction and curated regulatory domains were added to the basal regulatory domain. *Known* and de novo motif identification was performed with HOMER 4.8[78].

**Comparative analysis of ChIP-seq data**. Bedtools at the Galaxy public server (https://usegalaxy.org) were used to compare and map data sets for different

transcription factors. Analyses were visually represented using IGV software[79], which is publicly available from the Broad Institute website (https://software.broadinstitute.org/software/igv/).

**Reporting summary**. Further information on research design is available in the Nature Research Reporting Summary linked to this article.

## Data availability

Sequencing data have been deposited in the GEO-NCBI database under accession code: GSE134039 for RNA-seq and GSE134034 for ChIP-seq. Summary tables of the RNA-seq and ChIP-seq analyses performed are available as Supplementary Data 1–4. Numerical source data corresponding to the graphs showing aggregated data in the figures are provided in a "Source Data" excel table. Source images corresponding to all individual specimens analyzed are available from "Mendeley Data" [https://data.mendeley.com/datasets/r774bxyf8d/2] under https://doi.org/10.17632/r774bxyf8d.1 [80]. Source data are provided with this paper.

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

## Acknowledgements

We thank members of the Torres group for stimulating discussions and suggestions. RNA probes were kindly provided by Marian Ros, Paola Bovolenta, and Jose Luis de la Pompa. We thank Keith Humphries and Hesham Sadek for providing the *Meis1* floxed mice and Susan Mackem for the *Hoxa6CreERT2* mice. We thank Alberto Benguría from the Genomics Unit, Manuel José Gómez from the Bioinformatics Unit, and Angeles Sanguino from the Pluripotent Cells Technology Unit. This study was supported by grant PGC2018-096486-B-I00 from the Spanish Ministerio de Ciencia e Innovación. The CNIC is supported by the Ministerio de Ciencia e Innovación and the Pro CNIC Foundation.

## Author contributions

Experimental procedures: I.D., G.G., S.T., A.B., and Y.G.; conceptualization and experimental design: I.D., M.T., and J.D.; data analysis: I.D. and M.T. Figure design: I.D. and M.T.; Manuscript writing: I.D. and M.T. Fund raising: M.T.

## Competing interests

The authors declare no competing interests.
