## [Peer Review File · Nature Communications]

Reviewers' Comments:

Reviewer #1:

Remarks to the Author:

In this study, Delgado et al examine the role of Meis TFs in initiation/patterning of the vertebrate limbs.

Overall, the experimental work is solid, well explained and provides key novel data sets for the field (in the form of RNA-seq and ChIP-seq).

In terms of the impact of this report, it should be noted that Meis TFs have been previously implicated in limb formation, but that such conclusions have not been verified by Meis knock-out analyses. By carrying out these analyses, the work by Delgado et al solidifies/validates previous work and therefore provides value to the field. However, the main novelty for the field would be to provide a mechanistic basis for Meis function in the limb. In this regard, the authors argue that Meis regulates expression of key limb initiation factors (e.g. fgf10) in cooperation with Tbx TFs and that Meis also regulates key limb AP patterning genes (e.g. hand2) - possibly in cooperation with Hoxd13. These conclusions would represent a key advancement in the field, but are not fully substantiated by the reported work and need to be addressed as follows:

1. The data demonstrating co-binding by Meis and Tbx at key limb regulators is convincing, but would be further enhanced by carrying out a de novo motif search at Meis/Tbx co-bound sites. Is the Meis:Tbx compound motif more prevalent at co-bound sites?
2. The deletion of Meis:Tbx:Hox10 co-bound elements is a key test of the authors mechanistic model, but Fig. 4D is not convincing. In particular, fgf10 staining in the 5' deletion appears reduced in the whole embryo - i.e tailbud expression is also reduced. (What do arrowheads point to?). Also, in the 14.5kb deletion, expression seems lost in the HL, but Tbx5 is not active in the HL. Is Tbx4 bound in this region? Overall, in situ hybridizations are not very quantitative and a different method should be used to demonstrate changes in fgf10 expression. This is particularly important since this effect appears transient and differences in staging of the embryos could have a significant effect.
3. The authors examine the role of Meis in AP patterning of the limb. By examining expression of genes known to act in AP patterning, they conclude that Meis acts upstream of Hand2. Based on ChIP data, they suggest that Meis may activate Hoxd13 expression and subsequently act with Hoxd13 to regulate Hand2. This is an exciting model, but it is not tested functionally. Unless this model is tested, the analysis of Meis' role in AP patterning does not provide much new information.

Minor points:

1. It appears that pknox2 is upregulated in Meis KO. Since pknox2 is closely related to Meis, it may compensate for loss of Meis. This should be discussed.
2. The figures would benefit from more arrows etc to highlight the features discussed in the text.

Reviewer #2:

Remarks to the Author:

Delgado and colleagues have generated and analyzed mouse embryos lacking both the Meis1 and Meis2 transcriptional regulators specifically in mouse limb buds. Their analysis provides the answers to longstanding open questions in developmental biology namely what is the role of these two key regulators during the onset of limb bud and proximo-distal limb axis development (compagnon paper). This is a nice study with important conclusions.

Their analysis of limb bud initiation shows that Meis 1/2 are required in a functionally redundant

manner for establishing the mesenchymal Fgf10 expression, which is the earliest signal known to initiate the Fgf signaling system that controls proximo-distal outgrowth. Inactivation of both Meis transcriptional regulators results in complete limb agenesis.

The authors also analyze hypomorphic limbs in which only 3 of the 4 Meis alleles are inactivated. This analysis establishes that Meis genes also are required for antero-posterior pre-patterning which in turn activates Shh signaling. This later discovery is rather unexpected and together the other results reveals the key functions of Meis 1 / 2 during limb bud initiation and in setting up the PD and AP limb axis patterning systems.

In revising their manuscript, the authors should address the following issues:

1. As a fraction of the RNA in situ analyses is done at stages when mutant limb buds look clearly smaller than wildtype controls, it is important to include cell death and proliferation analyses of Wt and mutant limb buds at E10.0 – E10.5. This is not only important with respect to the RNA expression analyses but will answer the question if progenitors are eliminated by apoptosis and/or if there is a proliferation arrest.

2. For transcriptome and ChIP-seq analyses it is standard to deposit the datasets in a repository and to include supplemental tables listing all differentially expressed genes (RNA-seq); called peaks and associated genes (ChIP-seq) with the manuscript.

3. From the methods it is clear that there were two replicants for the RNA-seq but only one replicate for the ChIP-seq analysis. More details must be included for the analysis of the transcriptome such as e.g. tables listing all DEGs, the genes up-and down regulated in both samples, FDR and so on.

Why was only one replicate done for the ChIP seq? How were the peaks called for the ChIP-seq analysis and associated to the candidate target gene (peaks in TAD or closest TSS)?

4. Comparative analysis of Meis, Tbx and Hox-binding sites: again the genes used to derive the diagrams shown in Fig. 4 A/B must be listed in supplementary tables and new datasets deposited.

5. The description of the data shown in Fig. 4 and the conclusions drawn from the bio-informatics analyses are at best indicative of direct interactions/cooperation as no follow experiments were done to establish direct interactions and functional relevance. Therefore the conclusion that the authors draw from this analysis (page 12, lines 280-282) are worded much too strong and need to be tuned down.

ADDITIONAL POINTS

6. The authors should check that the description of the results matches the order of the panels in the Figures (see e.g. Figure 2).

7. There are some spelling mistakes in the text. In Figure 5H, the term "Bended Z" should be changed to "Bent Z".

Reviewer #3:

Remarks to the Author:

Interesting phenotype generated by conditional double KO of Meis1/2. This information has been missing from the literature and is interesting. It would be helpful if skeletal preps from later time points are included – or perhaps the embryos die? This information should be included in presentation of phenotype.

For the 71 downregulated and 355 upregulated in RNA-seq analyses, statistics are not included in

the presentation of this data. For instance, what was cut-off for consideration as up- or down-regulated and how was statistical significance calculated to determine this? 2 samples for WT and 3 from mutant may not be sufficient to do acceptably rigorous statistics – and 2 counts per million seems highly unlikely to be sufficient for 'detectable' to be assigned; there is no description provided regarding how statistically significant detection of expression or statistical differences between groups were determined. This information can only be assessed with information supported by rigorous statistics – and presentation of statistics and metrics used to support interpretations needs to be added.

In addition to previous concern, in text, when gene expression differences are mentioned, fold-changes must be stated (i.e. *fgf8*, *fgf10*, *lef1*, *alx1*, 2, 3, *shox2*, *tbx1*, *hoxa13*, *pknox2*, etc...). Some indication of what the most significant changes in expression are in the overall analyses (within or out of the sets of genes mentioned) should also be included in a visible way in figure or text.

Figure 3. It is unclear to this reviewer how 6811 (FL) and 6010 (HL) peaks result in a higher, 7202, consensus set identified. What were statistics that allowed shared peaks (5803) to be defined?

This reviewer is confused regarding how the preference for binding sites can be the intron (50-500 bp downstream of start site) and yet ~80% of genes in the region of the peaks are near 2 genes.

The foundation of Figure 1 that *Fgf* expression is affected, which is strongly supported, is not supported by ChIP-seq data; weakening assertion that *Meis* interacts with *Tbx* to directly regulate expression. The transgenic test of potential enhancers in Figure 4 does not lend confidence to this assertion. Taken together, the assertion that *Fgf* is directly regulated is weak. Concluding that *Meis*, *Tbx* and *Hox* cooperative regulate is not supported and should be removed (line 280) – the evidence is insufficient to support this claim.

It is critical that the authors not rely on published ChIP-seq data on *Hox* binding to claim they are specific binding sites for particular *Hox* protein(s). As an example, that it is reported that *Hoxc10* binds at these sites is not sufficient to label this as a *Hoxc10* binding site as distinguished differentially from other *Hox* proteins also potentially binding. Particularly given the inability to explain phenotypes in the context of a given *Hox* genes or paralous set of genes. Discussion with regard to specific *Hox* binding sites is not accurate if it is not supported by more than single, non-comparative kind of data set. This discussion must undergo major modifications using more rigorous assumptions in using published data as a context for interpretation. This arise in several areas of the manuscript.

A previously described *Tbx*-binding site is referred to on line 220 – this is not referenced and, if available, could have been used for previous discussion of potential for *Tbx/Meis* interaction.

Figure 6 and 7 refer to the same point and should be presented together. The authors focus on *Hoxd13* and it has been shown to have a relatively minor role in *Hand2* expression, but given *Hox9* mutants lead to an absence of *Hand2* expression and a phenotype that supports that (unlike *Hoxd13* or other *Hox12/13* genes), discussion and presentation of this data is a bit misleading– given the results with *Hoxd9* and *Hond2* expression in HL where *Meis* is totally absent, the authors cannot state that *Meis* regulates *Gli*, *Hand*, *Hox* (line 357) with this collective data. Altogether, this section of the data is very difficult to interpret, but the authors do not support their interpretation with the data shown.

We thank the reviewers for their constructive comments on our manuscript. We have thoroughly considered their requests in revising the manuscript.

Data availability for review:

The GEO reference for the RNAseq is GSE134039 and ChIPseq data is GSE134034. The access token for reviewers is **glsreagexdqtsl**.

Supplementary excel files containing the relevant results of RNAseq and ChIPseq have been included in the submission.

A web browser version of the ChIPseq analysis is available here:

<http://genome.ucsc.edu/s/mtorres%40cnic.es/Meis%20ChIPseq%20E10.5%20buds>

The complete set of source images corresponding to all individual specimens analyzed are available to reviewers in "Mendeley Data" in this link:

<https://data.mendeley.com/datasets/r774bxyf8d/draft?a=88c458f0-aa65-4bd3-82f0-b5f74a1a3cd5>

Numerical source data corresponding to the graphs in the figures are provided in an excel file with the submission

Reviewers' comments:

Reviewer #1 (Remarks to the Author):

In this study, Delgado et al examine the role of Meis TFs in initiation/patterning of the vertebrate limbs.

Overall, the experimental work is solid, well explained and provides key novel data sets for the field (in the form of RNA-seq and ChIP-seq).

In terms of the impact of this report, it should be noted that Meis TFs have been previously implicated in limb formation, but that such conclusions have not been verified by Meis knock-out analyses. By carrying out these analyses, the work by Delgado et al solidifies/validates previous work and therefore provides value to the field. However, the main novelty for the field would be to provide a mechanistic basis for Meis function in the limb. In this regard, the authors argue that Meis regulates expression of key limb initiation factors (e.g. fgf10) in cooperation with Tbx TFs and that Meis also regulates key limb AP patterning genes (e.g. hand2) - possibly in cooperation with Hoxd13. These conclusions would represent a key advancement in the field, but are not fully substantiated by the reported work and need to be addressed as follows:

1. The data demonstrating co-binding by Meis and Tbx at key limb regulators is

convincing, but would be further enhanced by carrying out a *de novo* motif search at Meis/Tbx co-bound sites. Is the Meis:Tbx compound motif more prevalent at co-bound sites?

We have now done both *de novo* and *known-motif* searches in Meis-only and Meis-Tbx5 common regions. We report the results in the new figure S6. The results show that, for *de novo* motifs, Tbx5 binding sites are found in 30% of the regions, with a p-value of 1^{-34} . In contrast, the Tbx5 motif was not significantly detected in the Meis-only peak set. When looking for known motifs, the Tbx5 motif was detected at p-value of 1^{-59} , while in the Meis-only set, it was identified at p-value 1^{-15} . These results reinforce the conclusion of the association of Meis and Tbx binding sites.

2. The deletion of Meis:Tbx:Hox10 co-bound elements is a key test of the authors mechanistic model, but Fig. 4D is not convincing. In particular, fgf10 staining in the 5' deletion appears reduced in the whole embryo - i.e tailbud expression is also reduced. (What do arrowheads point to?).

Arrowheads point to the regions with Fgf10 expression (eye, FL and tailbud). Now we have replaced these by asterisks and describe what they indicate in the Figure legend. All embryos with 5'-Δ0.4 and 3'-Δ0.4 deletions analyzed by *in situ* hybridization show a normal expression in the eye and tailbud, which shows that the reduction in limb bud expression is limb-specific. In contrast, the 14.5kb deletion shows a reduction of expression as well in the other regions, indicating that this region may contain additional general enhancers of *Fgf10* expression. We now explain these observations in detail in the revised manuscript.

Also, in the 14.5kb deletion, expression seems lost in the HL, but Tbx5 is not active in the HL. Is Tbx4 bound in this region?

At the stages analyzed HL Fgf10 expression is not detectable, so we do not have evidence of affection of the hindlimb and our conclusions can only be applied to FLs. Tbx4 was not detected in the only published ChIP-seq study for this protein in hindlimbs (Jain et al., Development 2018). We cannot exclude the possibility that the common binding described only takes place in FLs, however, the sensitivity of the Tbx4 ChIP-seq study was very much reduced compared to that of the Tbx5 ChIP-seq (4780 Tbx4 vs. 10273 Tbx5 peaks) and therefore, it is also likely that the peak did not appear due to the lower sensitivity of the assay. We have now assayed the *Fgf10* deletions for genetic interaction with Meis kos (see below) and the results indicate the functionality of the regulatory regions identified in both fore- and hindlimbs.

Overall, *in situ* hybridizations are not very quantitative and a different method should be used to demonstrate changes in fgf10 expression. This is particularly important since this effect appears transient and differences in staging of the embryos could have a significant effect.

qPCR was performed and included in the revised manuscript (Figure 5E). Both, *in situ* hybridization and qPCR KO embryos were somite-staged and compared with the same somite-staged WT embryo; therefore, staging should not represent a confounding factor. The qPCR analysis confirms the observations from whole-mount *in situ* hybridization.

Furthermore, we went a step ahead and generated stable mouse lines carrying the 5'0.4kb and 14.5kb deletions. In embryos homozygous for each of the deletions, the reduction observed in *Fgf10* expression was transient. By E10.5, *Fgf10* expression has recovered and limb development proceeds normally. This result indicates redundancy with other *Fgf10* enhancers for limb initiation. To determine whether the deletions constitute a background sensitized to Meis loss, we crossed the deletions to Meis KOs and found that the combination of the deletions with Meis mutations is interactive and exacerbates the Meis loss of function defects. These results demonstrate the relevance of the elements identified and their functional interaction with Meis factors during limb initiation. Unexpectedly, these experiments showed a strong rescue of the stylopod length, which shows that weakening distalization, rescues the proximal program. This observation supports the previously proposed two-signal model for limb PD patterning.

3. The authors examine the role of Meis in AP patterning of the limb. By examining expression of genes known to act in AP patterning, they conclude that Meis acts upstream of *Hand2*. Based on ChIP data, they suggest that Meis may activate *Hoxd13* expression and subsequently act with *Hoxd13* to regulate *Hand2*. This is an exciting model, but it is not tested functionally. Unless this model is tested, the analysis of Meis' role in AP patterning does not provide much new information.

To strengthen the study, we studied the candidate regulatory regions and identified a distal region in the *Hand2* locus with Meis and *Hoxd13* binding sites (Figure 6F). This region also contains an enhancer that had previously been characterized for limb activity in transgenics (Monti et al., PLOS Computational Biology 2017) and overlaps Meis and *Hoxd13* binding sites (Figure 6F). Here, we explored the relevance of this region *in vivo* and *in locus* by CRISPR-Cas9 deletion. We found that the deletion reduces *Hand2* limb bud expression in heterozygosis and completely eliminates it in homozygosis, while it does not affect *Hand2* expression in other embryonic regions (Figure 6G).

We also deleted the Meis binding region in the *Hoxd13* locus, however, this binding region comprises the whole *Hoxd13* exon2 and, therefore, its deletion interrupts a large part of the transcriptional unit, making it difficult to interpret the results. While, in this case, we could not demonstrate the relevance of the region bound by Meis, the absence of *Hoxd13* expression in the Meis mutants indicates that either directly or indirectly, *Hoxd13* requires Meis activity to get activated in the early posterior limb bud, which fits the model proposed.

Minor points:

1. It appears that *pknx2* is upregulated in Meis KO. Since *pknx2* is closely related to Meis, it may compensate for loss of Meis. This should be discussed.

Although closely related, functional evidence does not support redundancy of the *pknx* (Prep) and Meis families. ChIPseq analyses show very different patterns of target sequence choice and genomic binding sites (Penkov et al. Cell Reports 2014). Some evidence from oncogenic activity suggests the two families antagonize rather than providing similar functions (Dardaei et al., PNAS 2014). In the limb bud, Meis is

expressed down to the zeugopod/autopod boundary, while *Pknox2* is expressed in the presumptive zeugopod region and its misexpression specifically affects the zeugopod (Zhou et al., 2013). In the absence of a triple *Meis1*, *Meis2*, *Pknox2* ko, we cannot rule out either antagonistic or cooperative functions with *Pknox2* (or neutrality), however, the strong phenotype associated to the complete elimination of *Meis1* and *Meis2* suggests the inability of other transcription factors to compensate for the loss of *Meis* function in the limb. We have included a comment on these aspects in the discussion as suggested by the reviewer.

2. The figures would benefit from more arrows etc to highlight the features discussed in the text.

We have improved the figures and labelled them further to facilitate comprehension.

Reviewer #2 (Remarks to the Author):

Delgado and colleagues have generated and analyzed mouse embryos lacking both the *Meis1* and *Meis2* transcriptional regulators specifically in mouse limb buds. Their analysis provides the answers to longstanding open questions in developmental biology namely what is the role of these two key regulators during the onset of limb bud and proximo-distal limb axis development (compagnion paper). This is a nice study with important conclusions.

Their analysis of limb bud initiation shows that *Meis 1/2* are required in a functionally redundant manner for establishing the mesenchymal *Fgf10* expression, which is the earliest signal known to initiate the *Fgf* signaling system that controls proximo-distal outgrowth. Inactivation of both *Meis* transcriptional regulators results in complete limb agenesis.

The authors also analyze hypomorphic limbs in which only 3 of the 4 *Meis* alleles are inactivated. This analysis establishes that *Meis* genes also are required for antero-posterior pre-patterning which in turn activates *Shh* signaling. This later discovery is rather unexpected and together the other results reveals the key functions of *Meis 1 / 2* during limb bud initiation and in setting up the PD and AP limb axis patterning systems.

In revising their manuscript, the authors should address the following issues:

1. As a fraction of the RNA in situ analyses is done at stages when mutant limb buds look clearly smaller than wildtype controls, it is important to include cell death and proliferation analyses of Wt and mutant limb buds at E10.0 – E10.5. This is not only important with respect to the RNA expression analyses but will answer the question if progenitors are eliminated by apoptosis and/or if there is a proliferation arrest. As requested, we have now determined cell death and proliferation patterns in mutant E10 limbs (Figure S2). We found no difference in proliferation in mutant versus WT

limbs. We found a clear increase of cell death in mutant FLs, however, due to the recombination pattern of Hoxb6CreER, in most cases AER and an incipient FL bud is formed; therefore, FLs do not represent the full elimination of Meis and it is possible that initial limb bud formation later regresses by a reduction in AER signals. In contrast, we found only a slight increase of cell death in the mesoderm of mutant HLs, which does not explain the limb agenesis in M1KO;M2KO embryos. The results are therefore compatible with limb agenesis rather than generation and subsequent elimination of limb precursors in the total absence of Meis function. We also observed an increase in cell death in the ectoderm that might be related to the loss of FGF signaling in M1KO;M2KO limbs, given that Fgf8;4 mutants show increase cell death in the AER and dorsal ectoderm (Sun et al.,2002).

2. For transcriptome and ChIP-seq analyses it is standard to deposit the datasets in a repository and to include supplemental tables listing all differentially expressed genes (RNA-seq); called peaks and associated genes (ChIP-seq) with the manuscript.

The GEO reference for the RNAseq is GSE134039 and ChIPseq data is GSE134034. The access token for reviewers is glsreagexdqjtsl.

We have included the data requested by the reviewer as supplementary data set 1 to 4.

3. From the methods it is clear that there were two replicants for the RNA-seq but only one replicate for the ChIP-seq analysis. More details must be included for the analysis of the transcriptome such as e.g. tables listing all DEGs, the genes up-and down regulated in both samples, FDR and so on.

We have extended the analysis to include 5 mutant and 6 wild type biological replicates for RNAseq. The RNAseq data requested have been included in supplementary dataset 1.

Why was only one replicate done for the ChIP seq?

Taking into account that we ChIPed for endogenous transcription factors without any tags or overexpression and that our target tissue is the initiating limb, there were strong limitations for the collection of sufficient material. Each replica required ~500 limb buds very precisely staged (36so). We therefore decided to do the ChIP independently on fore- and hind-limbs, thereby maximizing the sample usability. Given the degree of overlap between the peak sets obtained from FLs and HLs (96% of HL peaks and 85% of FL peaks), despite known differences between them and the delay of HL development with respect to FLs, we believe the ChIPseq result are highly reliable.

How were the peaks called for the ChIP-seq analysis and associated to the candidate target gene (peaks in TAD or closest TSS)?

We have used the standard conditions suggested by the GREAT software (MacLean et al Nature Biotechnology 2010). Briefly, the software assigns peaks to genes by assigning a basal regulatory domain to each gene by extending from 5000bp upstream

the transcriptional start site to 1000bp downstream. The regulatory domain is extended in both directions to the nearest gene's basal regulatory domain but no more than 1000kb from the TSS in each direction and curated regulatory domains (experimentally mapped regulatory regions) are added as well to the basal regulatory domain. We have now included the explanation of these two aspects in the materials and methods section.

4. Comparative analysis of Meis, Tbx and Hox-binding sites: again the genes used to derive the diagrams shown in Fig. 4 A/B must be listed in supplementary tables and new datasets deposited.

The diagrams in 4A,B do not refer to genes but to binding sites (peaks). The list of Meis peaks is now provided in supplementary dataset 2. The list of binding sites used for the comparison with TBX5 and HoxC10 are published in Nemec et al., Development 2017 and Jain et al., Development 2018. The overlap was calculated by determining those peaks whose summits were closer than 100 bp, as mentioned in the text. We now provide in supplementary dataset 4 the bed tracks that include the detected overlaps.

5. The description of the data shown in Fig. 4 and the conclusions drawn from the bioinformatics analyses are at best indicative of direct interactions/cooperation as no follow experiments were done to establish direct interactions and functional relevance. Therefore the conclusion that the authors draw from this analysis (page 12, lines 280-282) are worded much too strong and need to be tuned down.

We have modified our statements throughout the manuscript, as requested by the reviewer.

ADDITIONAL POINTS

6. The authors should check that the description of the results matches the order of the panels in the Figures (see e.g. Figure 2).

We have reviewed and modified the order accordingly

7. There are some spelling mistakes in the text. In Figure 5H, the term "Bended Z" should be changed to "Bent Z".

Thanks for pointing it out

Reviewer #3 (Remarks to the Author):

Interesting phenotype generated by conditional double KO of Meis1/2. This information has been missing from the literature and is interesting. It would be helpful if skeletal preps from later time points are included – or perhaps the embryos die? This information should be included in presentation of phenotype.

Unfortunately we do not have access to later stages, as Double KO embryos start dying from E13.5 onwards due to hematopoietic defects.

For the 71 downregulated and 355 upregulated in RNA-seq analyses, statistics are not included in the presentation of this data. For instance, what was cut-off for consideration as up- or down-regulated and how was statistical significance calculated to determine this? 2 samples for WT and 3 from mutant may not be sufficient to do acceptably rigorous statistics – and 2 counts per million seems highly unlikely to be sufficient for ‘detectable’ to be assigned; there is no description provided regarding how statistically significant detection of expression or statistical differences between groups were determined. This information can only be assessed with information supported by rigorous statistics – and presentation of statistics and metrics used to support interpretations needs to be added.

RNAseq analysis now includes 6 WT limb samples and 5 M1KO;M2KO limbs, statistical analyses are described in materials and methods and the statistics for each detected gene are included in a full list of detected genes (Supplementary dataset 1).

In addition to previous concern, in text, when gene expression differences are mentioned, fold-changes must be stated (i.e. *fgf8*, *fgf10*, *lef1*, *alx1*, 2, 3, *shox2*, *tbx1*, *hoxa13*, *pknox2*, etc...). Some indication of what the most significant changes in expression are in the overall analyses (within or out of the sets of genes mentioned) should also be included in a visible way in figure or text.

Besides the Supplementary dataset 1 including all detected genes with their expression levels and statistics, we now present a volcano plot, where fold change and significance can be seen for the whole population and in which we have highlighted a set of relevant limb patterning genes.

Figure 3. It is unclear to this reviewer how 6811 (FL) and 6010 (HL) peaks result in a higher, 7202, consensus set identified. What were statistics that allowed shared peaks (5803) to be defined?

The consensus peak set contains the unified list of all non-redundant peaks identified and results from the sum of the shared peaks, plus the HL-specific genes, plus the FL-specific genes. The shared peaks were identified by detecting the peaks independently and determining their overlap between the two sets (summits closer than 100 bp). The cutoff value for the FL-only and HL-only peaks was $p \leq 0.0005$, while the cutoff for the shared peaks was $p \leq 0.0005$ for the combined occurrence of the two overlapping peaks. These details are now provided in the Materials and Methods section.

This reviewer is confused regarding how the preference for binding sites can be the intron (50-500 bp downstream of start site) and yet ~80% of genes in the region of the peaks are near 2 genes.

We are sorry that the units were not labelled in the graph showing the distance and orientation with respect to TSS. This graph is in kb, not in bp (fixed now). Therefore,

there is plenty flexibility for the peaks to be in regions that are rich in genes, especially since they are not particularly prone to appear near TSS.

The foundation of Figure 1 that Fgf expression is affected, which is strongly supported, is not supported by ChIP-seq data weakening assertion that Meis interacts with Tbx to directly regulate expression. The transgenic test of potential enhancers in Figure 4 does not lend confidence to this assertion. Taken together, the assertion that Fgf is directly regulated is weak.

In Figure 1 we show that both Fgf8 and Fgf10 are affected by Meis loss. The ChIPseq analysis shows binding of Meis to Fgf10 regulatory regions but not to those of Fgf8. We thus proposed a possible direct regulation of Fgf10 and indirect regulation of Fgf8 by Fgf10 (known from the literature). The results of the deletion of the Fgf10 regulatory regions have now been strengthened by quantitative PCR analysis of Fgf10 mRNA expression. Furthermore, we now show genetic interaction between Meis and the Fgf10 deletions. These results undoubtedly link functionally Meis to Fgf10 activation. Nonetheless, we have not formally demonstrated that it is the direct binding of Meis what regulates the identified enhancers and have now modulated the statements to reflect the limitations of the study, as suggested by the reviewer

Concluding that Meis, Tbx and Hox cooperative regulate is not supported and should be removed (line 280) – the evidence is insufficient to support this claim.

Again, it is true that our study does not demonstrate molecular cooperation between these factors, however, the genetic and expression analyses show the convergence of the action of the factors on the activation of key genes for limb patterning. We have now modified our statements accordingly.

It is critical that the authors not rely on published ChIP-seq data on Hox binding to claim they are specific binding sites for particular Hox protein(s). As an example, that it is reported that Hoxc10 binds at these sites is not sufficient to label this as a Hoxc10 binding site as distinguished differentially from other Hox proteins also potentially binding. Particularly given the inability to explain phenotypes in the context of a given Hox genes or paralous set of genes. Discussion with regard to specific Hox binding sites is not accurate if it is not supported by more than single, non-comparative kind of data set. This discussion must undergo major modifications using more rigorous assumptions in using published data as a context for interpretation. This arise in several areas of the manuscript.

We agree 100% with the reviewer. We do not think or claim the Hoxc10 binding sites are specific for Hoxc10. In fact, we put forward the consideration that they represent generic Hox-binding sites potentially bound by several paralog groups. This is our statement in pg 8: “These results suggest that most Hoxc10 binding sites are general limb-bud Hox target sequences and are not HL-specific“. We have revised the manuscript and think our arguments are valid. Using published data sets for meta-analysis is a common practice in today’s Biology, so we find it legitimate to use published ChIPseq data.

A previously described Tbx-binding site is referred to on line 220 – this is not referenced and, if available, could have been used for previous discussion of potential for Tbx/Meis interaction.

We have now included this reference. This study identified Tbx5 binding sites in a cardiomyocyte cell line. As mentioned in the discussion, Meis binding sequences are found in the binding sites of Tbx5 in cardiac tissues which strengthens the idea of a functional convergence of these factors during patterning different organs.

Figure 6 and 7 refer to the same point and should be presented together. The authors focus on *Hoxd13* and it has been shown to have a relatively minor role in *Hand2* expression, but given *Hox9* mutants lead to an absence of *Hand2* expression and a phenotype that supports that (unlike *Hoxd13* or other *Hox12/13* genes), discussion and presentation of this data is a bit misleading— given the results with *Hoxd9* and *Hand2* expression in HL where Meis is totally absent, the authors cannot state that Meis regulates *Gli*, *Hand*, *Hox* (line 357) with this collective data. Altogether, this section of the data is very difficult to interpret, but the authors do not support their interpretation with the data shown.

We have merged Figures 6 and 7, as requested. AP pre-patterning is indeed very different between forelimbs and hindlimbs. The role of *Hox9* paralogs in limb AP pre-patterning, mentioned by the reviewer, is indeed restricted to forelimbs and does not apply to hindlimbs. Hindlimbs are normal in compound *Hox9* paralog mutants (Xu and Wellik, PNAS 2011). The elimination of *Hoxd* 5' genes (d10-13) results in limbs that do not activate *Shh* (Tarchini et al., Nature 2006) and ectopic activation of *Hoxd11-13* leads to ectopic *Shh* expression (Zakany et al, Science 2004). Furthermore, *Hoxd13* binds and regulates *Hand2* and *Shh* regulatory regions and its ectopic expression leads to *Hand2* and *Shh* ectopic activation (Galli et al Plos Genetics 2010; Salsi et al Dev. Biol 2008). It is true that *Hoxd13* ko mice do not show AP patterning defects but, given the accumulated evidence, this is most likely due to redundancy with other 5' *Hox* genes co-expressed in the posterior limb bud. Our new result showing that the deletion of a *Hand2* regulatory region that contains Meis and *Hoxd13* binding sites completely and specifically eliminates limb bud *Hand2* expression, strengthens the proposed pathway. Nonetheless, the reviewer is correct in that our findings apply mostly to forelimbs and, although we found a similar functional outcome in hindlimbs, the specific molecular pathway is different in HLs and remains to be elucidated. We have now corrected the text to make these points clear and to specify that the role of *Hoxd13* is likely redundant with that of other *Hoxd* 5' genes.

Reviewers' Comments:

Reviewer #1:

Remarks to the Author:

My main concerns have been addressed. A few minor things to improve the manuscript:

1. There is still some confusion regarding the meaning of arrowheads, asterisks etc in some figures. E.g., in Fig. 5D, the legend and the text ascribe different purpose to asterisks and arrowheads (limb bud vs non-limb bud staining) and it is not clear if arrowheads point to FLs, as stated in legend, or to HLs.
2. There are still some instances of misspellings. E.g. heterocycgocity instead of heterozygosity
3. In the model in Fig. 7, it is not clear if the right panel refers to FL, HL or a combination of both.

Reviewer #2:

Remarks to the Author:

In revising their manuscript, Delgado et al. have made considerable efforts to address the criticisms raised by the three reviewers and generated additional data requested. As stated by this reviewers in the original review, the genetic and molecular analysis of Meis functions, identification of candidate target genes and potential interactions with other transcription factors add important novel information and understanding to coordinated PD and AP limb bud axes development that increases the novelty and general importance of this manuscript.

However, at this stage there is one major shortfall and that is the manuscript itself. It lacks a clear "red thread" and there is multiple issues with both text and figures.

What follows is a list of the major issues requiring attention

1. Organization of the manuscript: the manuscript would gain clarity from a more logical flow beginning e.g. with generation of the mice (Fig. 1), description of the limb phenotypes (current Fig. 4), RNA seq and ChIP-seq analysis (current Fig. 3), followed by the in-depth molecular analysis (current Fig. 2, 5, 6). The model figure 7 should be discussed in detail and not just mentioned in the final sentence of the discussion.
2. The text is in significant parts difficult to read and comprehend, partly due to very long run-on sentences. The following are select examples illustrate the problems with the text and some issues with figures.

Lines 154 – 161:

"...20-to-40% PD shorter than normal"

Should read:

"20% - 40% reduced along the PD axis in comparison to wild-type controls"

Lines 163 – 165:

It is still not clear how the consensus set of 7202 peaks from FLs and HLs was compiled and how it differs from the 5803 peaks shared between FLs and HLs. This must be clarified by more detailed description.

Line 188:

Figure 3e should be indicated in the text.

Line 198 – 199:

Do the authors mean (?):

"...classification of Meis binding sites into two groups, the ones shared with Tbx5 and the ones that are Meis-specific"

Lines 216 onward:

It should be clearly stated at the beginning of describing these results that the Hoxc10 dataset was used for mapping Hox binding sites (or similar). From then on only the term Hox binding sites/interactions should be used. The current text is confusing with respect to this issue (was pointed out by one of the reviewers previously).

Line 267 – 268

Starting with "These results indicate..."

The authors should clearly state why the deletion of the Fgf 10 enhancer regions does not result in a limb phenotype as long as the Meis genes are not inactivated. The most likely explanation for this is compensation in cis due to the presence of additional enhancers that also interact with Meis transcription factors (i.e. redundancy in cis or not?)

Line 312

What is a "specular appearance"?

Page 13

Figures are indicated wrongly throughout this page:

Figure S6 should be Figure S4

Figure 7 should be Figure 6

Lines 333 – 335

This sentence needs clarification.

Discussion

Figure 7 should be described in more detail.

Misspelling of heterozygous, heterozygosity

Figures

Figure 3b:

It is hard to read the gene names superimposed on the volcano blot. Can a heat map (wt v. mut) be added for the relevant genes?

Figure 4g:

The different shades of grey for wt and mutant genotypes do not show well on screen.

Figure 5d:

The in situs are very nice, but the panels are much too small. Suggestion: make Fig. 5d a separate Figure (maybe together with the analysis of the limb skeletons).

Figure 7:

Needs detailed description.

Supplementary Fig 2:

It is difficult to see the RED TUNEL staining in the fluorescent images. Maybe include individual channels in addition to the overlay (?).

Reviewer #3:

Remarks to the Author:

I think the authors have addressed the comments/questions posed by this reviewer and the resulting work will be of high interest to the field.

One minor comment is the response of the embryos dying at E13.5. The authors responded with this answer, but I did not note that it was included somewhere in the manuscript and think this is relevant information to include somewhere if it is not incorporated currently.

Response to Reviewers

We thank the reviewers for their constructive comments

Reviewer #1 (Remarks to the Author):

My main concerns have been addressed. A few minor things to improve the manuscript:

1. There is still some confusion regarding the meaning of arrowheads, asterisks etc in some figures. E.g., in Fig. 5D, the legend and the text ascribe different purpose to asterisks and arrowheads (limb bud vs non-limb bud staining) and it is not clear if arrowheads point to FLs, as stated in legend, or to HLs.

R. We have corrected and extended the description of Figure labelling

2. There are still some instances of misspellings. E.g. heterocycocity instead of heterozygosity.

R. Corrected

3. In the model in Fig. 7, it is not clear if the right panel refers to FL, HL or a combination of both.

R. Corrected

Reviewer #2 (Remarks to the Author):

In revising their manuscript, Delgado et al. have made considerable efforts to address the criticisms raised by the three reviewers and generated additional data requested. As stated by this reviewers in the original review, the genetic and molecular analysis of Meis functions, identification of candidate target genes and potential interactions with other transcription factors add important novel information and understanding to coordinated PD and AP limb bud axes development that increases the novelty and general importance of this manuscript.

However, at this stage there is one major shortfall and that is the manuscript itself. It lacks a clear "red thread" and there is multiple issues with both text and figures.

What follows is a list of the major issues requiring attention

1. Organization of the manuscript: the manuscript would gain clarity from a more logical flow beginning e.g. with generation of the mice (Fig. 1), description of the limb phenotypes (current Fig. 4), RNA seq and ChIP-seq analysis (current Fig. 3), followed by the in-depth molecular analysis (current Fig. 2, 5, 6). The model figure 7 should be discussed in detail and not just mentioned in the final sentence of the discussion.

R. We have reorganized the flow of the manuscript as suggested by the reviewer.

2. The text is in significant parts difficult to read and comprehend, partly due to very long run-on sentences. The following are select examples illustrate the problems with the text and some issues with figures.

R. We have revised the text trying to improve readability.

Lines 154 – 161:

"...20-to-40% PD shorter than normal"

Should read:

"20% - 40% reduced along the PD axis in comparison to wild-type controls"

R. Corrected

Lines 163 – 165:

It is still not clear how the consensus set of 7202 peaks from FLs and HLs was compiled and how it differs from the 5803 peaks shared between FLs and HLs. This must be clarified by more detailed description.

R. We have now explained it better in the manuscript.

Line 188:

Figure 3e should be indicated in the text.

R. Corrected

Line 198 – 199:

Do the authors mean (?):

"...classification of Meis binding sites into two groups, the ones shared with Tbx5 and the ones that are Meis-specific"

R. Yes, this is what we meant. We corrected the text to make this point clear

Lines 216 onward:

It should be clearly stated at the beginning of describing these results that the Hoxc10 dataset was used for mapping Hox binding sites (or similar). From then on only the term Hox binding sites/interactions should be used. The current text is confusing with respect to this issue (was pointed out by one of the reviewers previously).

R. We have corrected the text accordingly

Line 267 – 268

Starting with “These results indicate...”

The authors should clearly state why the deletion of the Fgf 10 enhancer regions does not result in a limb phenotype as long as the Meis genes are not inactivated. The most likely explanation for this is compensation in cis due to the presence of additional enhancers that also interact with Meis transcription factors (i.e. redundancy in cis or not?)

R. We have made these points clear in both the Results and the Discussion sections

Line 312

What is a “specular appearance”?

R. That the posterior side of the expression domain resembles the specular image of the anterior side of the expression domain. We have now rephrased this part for a better explanation

Page 13

Figures are indicated wrongly throughout this page:

Figure S6 should be Figure S4

Figure 7 should be Figure 6

R. Corrected

Lines 333 – 335

This sentence needs clarification.

R. Corrected

Discussion

Figure 7 should be described in more detail.

R. Done (now, Figure 8)

Misspelling of heterozygous, heterozygosity

R. Corrected

Figures

Figure 3b:

It is hard to read the gene names superimposed on the volcano blot. Can a heat map (wt v. mut) be added for the relevant genes?

R. We have added a Heat map to the Figure

Figure 4g:

The different shades of grey for wt and mutant genotypes do not show well on screen.

R. We have replaced the colors for better identification

Figure 5d:

The in situs are very nice, but the panels are much too small. Suggestion: make Fig. 5d a separate Figure (maybe together with the analysis of the limb skeletons).

R. We have done as suggested by the reviewer

Figure 7:

Needs detailed description.

R. We now described the figure in the legend and in the Discussion

Supplementary Fig 2:

It is difficult to see the RED TUNEL staining in the fluorescent images. Maybe include individual channels in addition to the overlay (?).

R. We now provided magnifications of the images, so that all channels are easily visualized

Reviewer #3 (Remarks to the Author):

I think the authors have addressed the comments/questions posed by this reviewer and the resulting work will be of high interest to the field.

One minor comment is the response of the embryos dying at E13.5. The authors responded with this answer, but I did not note that it was included somewhere in the manuscript and think this is relevant information to include somewhere if it is not incorporated currently.

R. We have added this information to the manuscript